



# Reproducible and Relocatable Regional Ocean Modelling: Fundamentals and practices

Jeff Polton[1], James Harle[1], Jason Holt[1], Anna Katavouta[1], Dale Partridge[2], Jenny Jardine[1],
Sarah Wakelin[1], Julia Rulent[1], Anthony Wise[1], Katherine Hutchinson[3], David Byrne[1],
Diego Bruciaferri[4], Enda O'Dea[4], Michela De Dominicis[1], Pierre Mathiot[3], Andrew Coward[1],
Andrew Yool[1], Julien Palmiéri[1], Gennadi Lessin[2], Claudia Gabriela Mayorga-Adame[1], Valérie Le
Guennec[1,5], Alex Arnold[4], and Clément Rousset[3]

[1]National Oceanography Centre, UK
[2]Plymouth Marine Laboratory, UK
[3]Laboratoire LOCEAN/IPSL, France
[4]Met Office, UK
[5]Pusan National University, South Korea

**Correspondence:** J.A.Polton (jelt@noc.ac.uk)

**Abstract.**

In response to an increasing demand for bespoke or tailored regional ocean modelling configurations, we outline fundamental principles and practices that can expedite the process to generate new configurations. The paper develops the principle of Reproducibility and advocates adherence by presenting benefits to the community and user. The elements to this principle are

reproducible workflows and standardised assessment, with additional effort over existing working practices being balanced against the added value generated. The paper then decomposes the complex build process, for a new regional ocean configuration, into stages and presents guidance, advice and insight on each component. This advice is compiled from across the user community, is presented in the context of NEMOv4, though aims to transcend NEMO version. Detail and region specific worked examples are linked in companion repositories and DOIs. The aim is to broaden the user community skill base, and to

accelerate development of new configurations in order to increase available time exploiting the configurations.

## 1 Introduction

There is internationally an increasing demand for simulations of the marine environment to deepen our understanding of the marine system and its sensitivities in a changing climate. High profile issues include marine hazards from storms (Harley et al.,

2022; Masselink et al., 2016), sea level rise (Fox-Kemper et al., 2021; Ponte et al., 2019), management of blue carbon resources and understanding the potential marine impacts of climate change mitigation interventions, such as Marine offshore renewable energy (Dorrell et al., 2022) and land use change (Felgate et al., 2021).



While global ocean modelling products and research activities are increasing in resolution and sophistication, they are still a long way from the scale and process representation required to deliver accurate information on the coastal ocean. There are a number of reasons why it is advantageous to configure bespoke regional models: though data catalogues like the Copernicus Marine Service (CMS[1]) (and others) are rich resources for regional and global marine data, these can not always satisfy all user requirements. Motivations need not always be about spatial resolution. For example, missing processes can be an important driver for building a new configuration (perhaps addressing a lack of integrated physics and biogeochemistry or a lack of tidal processes in the "off the shelf" catalogue products). Alternatively bespoke model outputs might be required (such as high frequency output for specific subregions or new metrics). So the key advantages of regional configurations over global include benefits associated with resolution enhancements, design flexibility and computational efficiency (to contain only the region and processes that matter and to not worry about degrading the solution in other regions). On the other hand, key disadvantages include the need for lateral boundary conditions (which can be hard to obtain and a potential source of error); the human resource required to configure and maintain multiple regional domains; and the lack of a common experience across a global community of coastal ocean modellers.

The configuration of a regional ocean model has traditionally been a one-off event taking many months and requiring many, often subtle, expert decisions. Consequently, descriptions of the set up (e.g. in the literature) are relatively limited or hard to reproduce in its entirety. The need to configure multiple regional models in many different seas around the world has led us to develop a systematic workflow where NEMO (the Nucleus for European Modelling of the Ocean[2], Madec and Team) regional ocean configurations could be more efficiently built, deployed and reproduced. This reproducible workflow is not intended to be the sole authority on regional configuration setup, or provide a turn-key or black-box solution, instead it is designed to provide a set of guidelines for modellers to follow in order to capitalize on the usability and interoperability of the resulting simulations. Indeed other modelling systems, such as MITgcm (Marshall et al., 1997), ROMS (Shchepetkin and McWilliams, 2005) and FVCOM (Chen et al., 2006), can also be readily configured for new regional applications. Each have their own strengths which are largely set according to the model system's development history and the user's familiarity with the system. For example ROMS was specifically designed as a Regional Ocean Model, and MITgcm has been a model of choice for numerous idealised processes studies. On the other hand, FVCOM's unstructured mesh has made it a popular research code choice for multiscale coastal hydrodynamics. The concepts presented here are intended to be broadly applicable to any modelling system, though the worked examples are implemented within NEMO.

NEMO is an ocean modelling framework underpinned by a consortium of 5 large European research, climate and operational centres. It is well supported as an international community modelling code and consequently is employed as the ocean component for 9 of the 35 widely used CMIP6 models[3] (Eyring et al., 2016) and is used in the CMS catalogue of freely available marine data products. As a regional model it benefits substantially from these investments of effort. However, its origins

---

[1]CMS:https://marine.copernicus.eu

[2]NEMO:http://www.nemo-ocean.eu

[3]CMIP6 homepage: [last accessed 27 Jul 2022]





in large research and operational centres (where teams focus on specific configurations) has led, quite naturally, to barriers to
NEMO being more widely used in regional applications.

In our experience, each research question addressed with a regional model configuration requires a subtly different workflow. Sometimes this would be the requirement of different forcing, sometimes the use of ensemble simulations, or sometimes different domain files. The intention here, therefore, is not to provide a *manual* on how to build a configuration but instead to share the concepts that need to be considered and practices that can be utilised when building a new configuration. This is in
part done with code examples. Note that the intention is not to create an automatic method for generating new configurations (e.g. Trotta et al., 2021); NEMO is a continually evolving code base with frequent releases and updates, so any turn-key solution would be quickly depreciated and less appropriate for cutting-edge scientific endeavours. Furthermore, since the process as a whole is complex and an unwelcome barrier to new starters, we have found it instructive to offer recipes that guide the user through the stages that need to be considered. Users are then encouraged to modify these recipes for their own bespoke purpose,
gaining insight through doing so, whilst simultaneously preserving the reproducibility documentation.

On this journey we have developed methodologies that reinforce the principle of reproducibility, that is fundamental to the scientific method. In particular these practices are aimed at making the use of large modelling frameworks more accessible. These concepts, and benefits, are discussed in Section 2. In Section 3 we step through the important considerations that can guide the construction of a new regional configuration. Model and configuration specific details are abstracted worked examples
in linked repositories. In Section 4 important considerations are described for a selection of modules that can expand the suite of process representation beyond the hydrodynamics (i.e. biogeochemistry, waves, nested domains, and ice processes). Finally discussion and conclusions are in Section 5. The manuscript is specifically targeted at the NEMO framework, with the hope of thereby making NEMO a more accessible framework for regional ocean modelling. However, the concepts (if not the details) are readily transferable to any regional modelling system.

## 2  Reproducibility: A fundamental principle, its implementation and sustainability

The scientific method requires reproducibility. However there is no defined level of documentation or code sharing required to meet this condition. In our discipline, code has always been available on request from authors, but with increasingly complex code bases, significant levels of expert knowledge is increasingly required to be able to compile and implement the code. The established modelling frameworks such as the MITgcm[4], ROMS [5] and NEMO[6] all have comprehensive documentation,
with large self-supporting user communities and online forums. Within the NEMO framework this support is invaluable for new users getting started and for community engagement with system development. However this alone can not deliver reproducibility, which arguably is minimally implemented within our community.

It is easy to understand how the additional time burden and potential loss of "intellectual property" might disincentivise an individual in making their science *too* easy to reproduce. Indeed the strategy for how one chooses to make their work

---

[4]MITgcm documentation: https://mitgcm.org/documentation/ [last accessed 14Jul22]

[5]ROMS documentations: https://www.myroms.org/wiki/Documentation_Portal [last accessed 14 Jul22]

[6]NEMO documentations: https://www.nemo-ocean.eu [last accessed 14 Jul22]

## Reproducibility

**Reproducible Workflows**
- Version control repositories
- Scripts / containerisation
- Recipes
- Accessible forcing data
- Attached DOIs

**Standardised Assessment**
- Common diagnostics
- Common code base

**User/Community Value**
- Publishing requirement
- Accelerate debugging & development
- Recognition for "non-standard" outputs
- Democratisation of skills
- Shared knowledge base

**Figure 1.** The principle of Reproducibility is delivered by Reproducible Workflows and Standardised Assessment but this ideal can only be maintained when its contribution is understood and Valued.

reproducible, or the level one must attain, is not prescribed, and perhaps nor should it be. However this lack of prescription, and non trivial amount of expert knowledge required to generate or reproduce simulations, can enhance the barrier to new adoption of modelling frameworks.

Beyond being a mandate, *reproducibility* offers clear benefits to the community. Reproducibility leads to enhanced efficiency with less time "reinventing the wheel" or consumed by software problems and more time dedicated to science discovery or
project deliverables. Reproducibility can lead to a democratisation of skills across a user base and an upskilling for individuals. It can accelerate debugging and therefore accelerate development. Enhanced levels of reproducibility in existing configurations make delivery of new working configurations a realistic prospect within smaller research projects, and furthermore makes the process more accessible for new regional modellers.

Furthermore with the progression towards increasingly automated and integrated systems (e.g. UK NERC digital strategy[7])
there will be an increasing demand for machine-capable reproducibility.

In recent years, and in response to increasing demand for new regional configurations, the authors sought to develop such practices. The concepts outlined here are intended to transcend code versions and even modelling frameworks. They are emergent rather than novel; born and distilled from experience and intending to borrow good working practices from the software development community. To be effective they must be memorable, even obvious.
There are three elements that have precipitated out of our work towards Reproducibility (and we are still on the journey). Two activities: Reproducible Workflows and Standardised Assessment, and a third element: Value. For the endeavour to be

---

[7]Digitally Enabled Environmental Science: NERC digitial strategy 2021-2030 [last accessed 1 Jul 2022]



sustainable, the additional activities must produce a recognisable Value, that exceeds the effort. Schematically summarised in Fig. 1, these are each addressed in the following.

## 2.1 Reproducible configurations

The first activity within the enhanced Reproducibility principle is Reproducible Workflows (Fig. 1). Reproducible relocatable regional modelling workflows already exist. Indeed established and alternative workflows should be reviewed and considered when choosing a template appropriate for a new project. Seminal examples include:

- A Structured and Unstructured grid Relocatable ocean platform for Forecasting (SURF[8]) (Trotta et al., 2016, 2021). Also using NEMO (and unstructured modelling) to rapidly build and deploy configurations for real time maritime disaster
response. The focus being on operational deployment and reliability. This necessitates a high level of automation and reliance on mature code versions.

- The NEMO nowcast framework[9]. This is a well documented collection of Python modules that can be used to build a software system to run the NEMO ocean model in a daily nowcast/forecast mode. NEMO nowcast has different, though complimentary, ambitions to this manuscript but are likely to be of interest to the reader.

- The Salish Sea MEOPAR Project Documentation[10]. This includes extensive documentation for a regionally specific NEMO configuration of the Canadian Salish Sea, which is deployed in various research projects (e.g. Soontiens and Allen, 2017).

Guided by existing examples and our own experiences, in this section the focus is on workflows that can enhance reproducibilty, whilst maintaining scientific flexibility.

### 2.1.1 Organised workflows

The key route to effective workflow reproducibility, and its benefits, is via systematic documentation.

Central to the structure advocated here is the use of:

i Version control repositories for modifications to the standard NEMO source code. We arbitrarily choose git and GitHub.

ii Scripts for configuring parts of the set-up that can be automated and labour saving. These also reside in the git reposito-
ries.

iii Recipes to describe the whole process.

It was found that the recipes, which make the whole process transparent from software installation through to assessment and analysis, were especially important in democratising the build process. Even though they took some time to document, the

---

[8]SURF: https://www.surf-platform.org/tutorial.php [last accessed 1Jul2022]
[9]NEMO nowcast: https://nemo-nowcast.readthedocs.io/en/latest/ [last accessed 1Jul2022]
[10]Salish Sea: https://salishsea-meopar-docs.readthedocs.io/en/latest/code-notes/salishsea-nemo/index.html [last accessed 1Jul2022]



benefit was immediate since multiple scientists and students could independently work on projects without continual reliance
on over burdened NEMO specialists.

Documenting the whole process in detail was important since the recipes form a template for subsequent configurations, for which the required modifications are hard to anticipate and could vary in nature from HPC architecture changes to alternative boundary forcing data.

We found the GitHub platform convenient for workflow management, since the code modifications, scripts and recipes
(the latter being in the form of associated wikis) could be co-located under one repository. We also found that the design of an "optimal" template repository was allusive since our various projects and experiments had subtly different requirements making a universal template unwieldy. We found, therefore, the most efficient approach to getting the benefits of an accelerated start on new projects was to clone and modify an existing project.

Excellent (and inspirational) examples of this workflow include the longstanding and extensive Canadian Salish Sea MEOPAR
Project Documentation[11]. In the UK the requirements have not been geographically focused and so led to an emphasis on building new relocatable configurations, starting with Lighthouse Reef in Belize[12] as a demonstrator. Subsequently this was iterated to build early versions of *SEAsia*[13] repository which in turn spawned *Caribbean*[14], which was modified to have scripts to auto build and run clean prescribed experiments using data recovery from remote storage (jasmin.ac.uk) and compute on remote HPC resource (archer.ac.uk). These documented experiences underpinned an ability to scale the number of new configurations,
spawning configurations in the Bay of Bengal and East Arabian Sea (*BoBEAS*[15]) and a number of other configurations listed in the appendix.

### 2.1.2 Containerisation

Containerisation presents a complimentary route to reproducible workflows that addresses the challenge of code portability between machines. The container provides a reproducible environment in which code, an ocean model in this instance, can be
developed and executed. Thus, removing the common constraints of library conflicts and bespoke installations on unique HPC systems. Through the use of containers, we can begin to construct an end-to-end scientific study, even including the pre/post processing tools used in a peer review publication. The use of container images has been gaining traction in academia over the last decade with several instances of their use in the geophysical sciences e.g. Hacker et al. (2017), Melton et al. (2020) and Cheng et al. (2022).

Container images contain pre-built applications together with their dependencies, such as specific versions of programming languages and libraries required to run the application. This image file is used by the container software on the host system to construct a runtime environment from which to run the application, providing an attractive and lightweight method of

---

[11]Salish Sea: https://salishsea-meopar-docs.readthedocs.io/en/latest/code-notes/salishsea-nemo/index.html [last accessed 1Jul2022]

[12]Lighthouse reef: https://pynemo.readthedocs.io/en/latest/examples.html#example-2-lighthouse-reef [last accessed 1Jul2022]

[13]SEAsia: doi:10.5281/zenodo.6483231

[14]Caribbean: doi:10.5281/zenodo.3228088

[15]BoBEAS: doi:10.5281/zenodo.4014837



virtualising scientific code. It provides a runtime environment that is independent of the host system, highly configurable and removes the setup and compilation issues potentially faced by the user.

The idea of moving towards full reproducibility makes it easier for peers to appraise these numerical scientific studies, and possibly build on them in future. Providing consistent compute environment containers will also benefit the development cycle of the code and increase its longevity. There are many software containers to choose from: Docker[16], Shifter[17], Charliecloud[18], RunC[19], Singularity[20], Podman[21], etc. Each offer their own advantages and compromises. Two that have gained a lot of traction within the scientific community over recent years are Docker (with a focus on cloud computing and local desktop deployment)

and Singularity (in the realm of HPC systems). Though this is not yet a routine part of our workflow, it has been an essential part of several successful projects and its use continues to be explored.

**Example: Docker**  Docker is an established containerisation software package that effectively streamlines the process to build, launch and manage containers. It originally required root priviledges and message passing was not trivial. However, getting Docker installed and running a demonstration NEMO container on new machines was so simple we were able

to deliver workshop training to run NEMO configurations on participants' consumer grade laptops. Docker abstracted all the challenges of participants' subtly different miscrosoft and mac software libraries into a single controlled build within a container (with a linux OS) that they could each install. This demonstration is made available through the *Belize_workshop*[22] configuration.

A more complex example was to build a Docker container with MPICH - a portable implementation of Message Passing

Interface (MPI) standard - to compile and run the NEMO ocean engine with parallel processing to run on Google Cloud for a much larger computational problem, with message passing between containers, in the Bay of Bengal and East Arabian Sea (*BoBEAS*[23]: with MPI Docker implementation). This ocean configuration is subsequently implemented in a coupled ocean-land-atmosphere regional suite (Castillo et al., 2022).

**Example: Singularity**  One of main strengths of Singularity is to provide a means of containerisation to the scientific com-

puting and HPC communities. It is increasingly gaining traction within the academic communities, with key motivatiors being: Singularity is an open source project; and Singularity can be run without root privileges on the host machine.

In a recent eCSE (www.archer2.ac.uk) project, CoNES[24], we developed a general method of containerising NEMO using Singularity. We provided automated recipes by which a user can build a Singularity Image File with their chosen version and components of the NEMO code. This immutable image file is then portable to a range of host systems. As part of

---

[16]Docker: docker.com [last accessed 20 Jul 2022]

[17]Shifter: https://github.com/NERSC/shifter [last accessed 20 Jul 2022]

[18]Charliecloud: https://hpc.github.io/charliecloud [last accessed 20 Jul 2022]

[19]RunC: https://github.com/opencontainers/runc [last accessed 20 Jul 2022]

[20]Singularity: Introduction [last accessed 20 Jul 2022]

[21]Podman: https://podman.io/ [last accessed 21 Jul 2022]

[22]Belize workshop: doi:10.5281/zenodo.6451433

[23]BoBEAS: doi:10.5281/zenodo.4014837

[24]CoNES: https://cones.readthedocs.io/en/latest/ [last accessed 20 Jul 2023]





the CoNES project the performance of the containerised code was compared against a natively compiled code on the
ARCHER2 HPC system (archer2.ac.uk). With minimal optimisation, the NEMO containers performed, on the whole,
within a few percent of the native code's runtimes (in some instances actually faster).

Developing workflows for research can be a complicated and iterative process, and even more so on a shared and some-
what rigid production environment. Containers provide a flexible working environment for development and production. As
demonstrated, they can be a useful tool for teaching, and remove barriers for new users, by removing the overhead of setting
up software in new environments and the challenges faced when attempting to adapt incompatible instructions to their bespoke
environment.

### 2.1.3   Accessibility of forcing and input data

Accessible input and forcing data are fundamental to (e.g. machine readable) reproducible workflows and merit some comment.
There are two issues here. Firstly, a working configuration can only be uniquely defined if it includes specification of the
external input data (e.g. bathymetry file, initial conditions) and any forcing data (e.g. meteorology, tides, rivers) as well as
complete details of the model parameters and code used. Therefore replicating the results is only possible with those same
inputs. At one level this appears to be an issue of semantics, but precise terminology here offers clarity on what is required in
order to satisfy the expected publishing requirement of reproducibility. In short, all the input data should be available.
The second, follow on, issue is to address how this requirement can be satisfied, when for large model simulations this level
of data storage might be problematic without advance planning.

Clearly adopting a recipe approach, or even a container approach, has much to offer here. Effective scripting with these
tools can decrease the expertise level required to reproduce a configuration build. For established community models, such as
NEMO, the resulting "configuration defining" material can be reduced to a number of scripts and a small collection of bespoke
files that specify and execute modifications to standard downloadable source code and datasets.

Similarly effective scripting can alleviate the data storage burden associated with making the forcing data available. Atmo-
spheric forcing, in NEMO for example, is specified via namelist definitions and is mediated through weights files that transform
the original data onto the target grid. Effective scripting can be used to download the raw forcing files and generate the weights
files. In this way, a configuration with a namelist that specifies particular forcing can be reproduced with reduced effort.
These recommendations require the input and forcing data to be openly available. Of course this is not always possible if
input data is commercially sensitive. This can be problematic to the scientific method, though data that has had some level of
processing can sometimes be made available to satisfy both privacy and reproducibility requirements.

In summary there are a number of forcings datasets that are required, which need to be available if the configuration is to
be reproducible. By way of example, for a regional operational type ocean model (e.g. Graham et al., 2018), forcings include:
rivers; lateral boundary conditions (time-varying and tidal harmonics, as appropriate); surface boundary conditions. Finally, in
addition, the bathymetry and grid configuration is required. All these data require a level of preprocessing to prepare them for
use. A pragmatic middle ground would be to include the preprocessing methods, from the point of externally available sources,





in the configuration along with a small sample of processed model-ready files. To ensure that (i) the model can be run for a short period with demonstration files (ii) forcings for long simulations can be replicated in the same way.

## 2.2 Standardised Assessment

The second activity within the enhanced Reproducibility principle is Standardised Assessment. The accuracy of output from any model configuration should be assessed by comparing it to equivalent observations. In the case of idealised configurations an assessment can be made against expected outcomes from theory or laboratory experiments. This is done to quantify how closely the model is able to simulate the reality it attempts to replicate. For forecasting models it is clear why this important: the accuracy of predictions can have significant impacts on the communities involved. For scientific applications it is equally important when simulating realistic regions as the scientist must have confidence in analyses, inferences and conclusions about the physical processes of interest. Error compensation may mean that improving the model with new, realistic, process degrades the comparison with observations. This is likely to be acceptable for scientific applications, but less so for operational cases. Note that this is a principle rather than a prescription, since the requirements will vary according to the modelling application. In this section we provide an outline of the key ideas behind the principle, highlighting the net benefits and advocating its importance. We consider there to be two elements to standardised assessment (see Fig. 1):

**A standardised framework** The framework prescribes templates for how different (class) objects should be structured, and requires all ingested data to be of a defined class. For example, all data are transformed into Xarray xarray objects with standardised dimensions and variable names according to its data type. This means that an equivalent assessment may be applied to data from different models (e.g. NEMO, ROMS, FVCOM, etc) and comparisons made to observations from any source (e.g. profile data from EN4 (Good et al., 2013) or World Ocean Database (Boyer et al., 2018a) sources).

**Common diagnostics** By defining classes to be built upon Xarray datasets, the powerful data handling capabilities are accessible to the framework. Furthermore, since all data loaded into the framework are of known type and properties, generic diagnostics can be written for each class which avoids the requirement for hardwired details relating to the data origin. The data source specific details are abstracted to the framework.

With this separation, contributors can more cleanly add to the diagnostics or the framework according to their interests and skills. This philosophy has been implemented, for example, in the COAsT[25] framework.

### 2.2.1 Benefits to the user

Standardised assessment workflows can benefit the user. Efficient workflows can accelerate the development process in highlighting how to iterate the configuration's tunable inputs. This practice may appear obvious, though in our experience tools for test driven development are far from standard in the oceanographic modelling community. The practice comes from software design where an extreme form of test driven development would be to write the tests before the application (see e.g. Beck,

---

[25]COAsT documentation: https://british-oceanographic-data-centre.github.io/COAsT/docs/ [last accessed 14 Jul 2022]





2002, for background). This extreme form may not be appropriate for ocean modelling, where the simulations are governed first by physical laws rather than skill, but the practice of having standardised assessment that can be easily executed (e.g. in a script form) could nonetheless accelerate refinement of tunable parameters that are otherwise unconstrained by the physical laws.

Progress in HPC performance produces simulations with increasingly larger volumes of data. These datasets increasingly require specialised tools to diagnose or manipulate. Standardised workflows for assessment can be built that abstracts aspects of the high performance computing data manipulation into generalised software libraries, and thereby jointly frees up science time whilst also increasing access to a broader range of scientific users. Furthermore, with community engagement in a standardised workflow, a broad range of specialist skills can each contribute their expertise to the mutual benefit of all the users.

### 2.2.2 Benefits to the scientific knowledge base

Ideally, Standardised Assessment is bound with the principle of Reproducibility and reproducible workflows, so that any shared configuration would include the scripts to generate it and also the means to generate a verification report. This practice would appear similar to the Copernicus Marine Service who provide Quality Information Documents with the catalogue (e.g. The Atlantic-European North West Shelf-Ocean Physics Reanalysis Quality Information Document[26]), though these would be entirely automatically generated.

Standardised assessment makes it easier for the community to assess simulations of the same domain, the full battery of results could be expected to appear as part of a published configuration (not necessarily peer-reviewed). This would allow for the quick and easy identification of the resultant impact of changes to the code, or alterations to parameterizations or boundary conditions, for certain pre-defined cornerstone metrics.

This transparency would relieve problems associated with selectively presenting only the most favourable outcomes when publishing in the peer review literature (under page count constraints).

### 2.2.3 Examples

The concept of standard or shareable verification, or assessment, tools is not novel. Indeed the concept is born from demonstrated successes with increased productivity (in more rapidly developing cumbersome NEMO simulations) and increased user engagement (using well written worked examples to develop NEMO and python skills). Many package examples exist, each with their specific motivations and specification. Notable mentions include:

- ESMValTool[27]: a community diagnostic and performance metrics tool for routine evaluation of Earth system models in CMIP.

---

[26]CMS AMM7 Multiyear reanalysis QuID [accessed 20May2022]

[27]ESMValTool: https://www.esmvaltool.org/index.html



- COAsT (Coastal Ocean Assessment Toolkit)[28]: a diagnostic and assessment toolbox targeted at kilometric scale regional models leaning heavily on the *Xarray* and *Dask* Python libraries (Hoyer and Hamman, 2017). The package brings simulations and observations into a common framework to facilitate assessment (see Byrne et al. (2022)).

- Pangeo[29]: a community project promoting open, reproducible, and scalable science providing documentation, developing and maintaining software, and deploying computing infrastructure to make scientific research and programming easier. Pangeo offers to scientists guides for accessing data and performing analysis using open source tools such as xarray, iris, dask, jupyter, and many other packages.

- nctoolkit[30]: a comprehensive and computationally efficient Python package for analysing and postprocessing netCDF data.

Taking example from the open source python user community, much of the scientific software written is underpinned by open source packages such as *NumPy*, for scientific computing (Harris et al., 2020); *Matplotlib*, for plotting (Hunter, 2007); *Xarray*, for manipulating multi-dimensional data, originally with a geoscientific focus (Hoyer and Hamman, 2017). Though these do not directly lead to standardised assessment tools it is important to highlight: a) their fundamental importance in underpinning any development of open source python scientific software development; b) they also serve as successful templates for how to coordinate, develop and maintain standardised assessment tools.

### 2.2.4 Cautions

Finally two notes of caution. The rise and fall of software stacks is strongly influenced by the ease at which it can be adapted (and kept up to date) and adopted (for new users, is the learning investment required offset by the expected gains?). The former is addressed by the aforementioned design choices, with the additional observation that (at least in a python environment) software appears to benefit from regular refreshing to update libraries and avoid obsolescence through depreciated code. The latter is aided with thorough documentation and *working* worked examples: worked examples accelerate user adoption and give insights into how the package is designed to be used.

The second note of caution is in "standardised assessments" being applied beyond the scope of their design specification or expected use. For example, erroneously inputting daily instantaneous temperatures when daily averages were expected would produced biased results. Or a more subtle example can be demonstrated when computing flux quantities as the product of two variables: combining 5-day averaged fields can miss the bolus interaction if the fields are correlated at at higher frequency (e.g. heat or volume transport calculations). To this end, the code would prioritise readability over efficiency and users would be encouraged to help improve the codebase.

---

[28]COAsT: documentation, DOI
[29]Pangeo: https://pangeo.io/
[30]nctoolkit:documentation [last accessed 20 Jul2 022]





### 2.3 Sustainability: a tension between cost and value

The above recommendations to enhance reproducibility have a fundamental problem: they require additional work from individuals.

Clearly a user can benefit from workflows and tools that others have created without adding to the knowledge base. This would be of no additional cost to themselves. A user could also choose to curate their code and notes in version controlled repositories. Except for some initial familiarisation with new tools and practices, this is not additional work. Furthermore they 305 would likely benefit from accelerated debugging particularly if help is sought.

However for the process to work, content must be created by people with appropriate expert knowledge and disseminated for the benefit of those seeking to upskill in this area. In some national laboratories or under large consortia programmes, roles can exist to support this community endeavor. For example *NumPy* has direct sponsorship and NEMO itself is sustained by significant support across an international consortium. However, with limited resource any activities not appropriately valued 310 will suffer neglect. A crucial aspect, therefore, is to appropriately recognise and reward the contributors for the value realised in shareable model configurations and assessment approaches. For example through well thought-out career pathways that acknowledge non-traditional science outputs, where peer-review is not appropriate. Output metrics could include repository page views, downloads or other esteem indicators. A practical recommendation towards this would be to normalise inclusion of non-traditional contributions on CVs, which could be facilitated by the adoption of narrative format CVs. However regardless 315 of CV format, the change would fundamentally require a culture shift towards valuing community contributions alongside traditional peer reviewed publications.

### 3 Practical considerations and worked examples in building a regional configuration

This section seeks to distill important considerations when building a new configuration. These insights are synthesised from a wide range of experience and expertise across the ocean modelling community and can help prioritise elements in a sequential 320 build plan.

The ordering is progressive, starting with prioritisation of the leading order processes for simulation, obtaining and building the code, proceeding through domain construction and then on to constructing external forcings. Each stage has a discussion highlighting the major options or considerations to be factored in the design. Each stage also has links to worked examples.

The technical detail in these worked examples is given in accompanying repositories. These are *SEAsia*[31], a 1/12 deg South 325 East Asia domain; *SANH*[32], a 1/12 deg South Asian domain; and *SEVERN-SWOT*[33], a 500m macro-tidal coastal domain. (See Fig. 2). Each configuration has subtly different requirements and subtly different challenges which influence configuration design. So we cite this range of examples to demonstrate a range of use case scenarios that might be instructive for the reader

---

[31]SEAsia: doi:10.5281/zenodo.6483231
[32]SANH: doi:10.5281/zenodo.6423211
[33]SEVERN-SWOT: doi:10.5281/zenodo.6469990

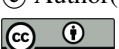



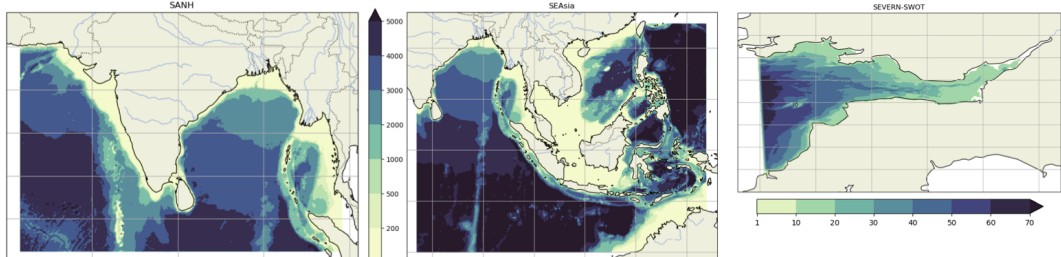

**Figure 2.** Bathymetry (m) from South East Asia, South Asia and Severn Estuary worked example configurations.

and to demonstrate the Reproducible workflows that we are advocating. (At the time of writing these configurations are all being actively developed. Though the releases pointed to herein are static and valid, they may be improved upon.)

## 3.1 Build planning and process prioritisation

The first stage consists of planning the new configuration and how to build it sequentially. The concept is to systematically increase the complexity of the processes represented, with associated efficacy testing. This is done in order to verify that the simulated processes behave as expected, and also to assist with error trapping should unexpected behaviours manifest. The first step, therefore, is to prioritise processes that need careful attention, or that dominate the system. For example, many simulations in the South China Sea may be dominated by tides, whereas simulations in the Mediterranean, or other inland seas, may be dominated by winds. Alternatively, for simulations in the northern Bay of Bengal, special consideration may be required to accurately represent the freshwater input and its seasonal variation. Similarly, in the case of models of the Arctic or Antarctic shelves, special care should be paid to properly represent sea ice and ice shelf processes. In practice, these priorities can not be set without consideration of the intended use of the model and will therefore be application specific. However, having formulated a priority list, experiments can be designed whereby separate processes can be tested to sequentially build complexity in the target configuration. This is particularly useful in light of the fact that the choice of the numerical techniques adopted to solve the governing equations will inevitably affect the realism and accuracy of many of the physical processes explicitly resolved or parameterised by the ocean model (e.g. Haidvogel, 1999; Griffies et al., 2000). Not all processes are separable, or easy to separate, and so a measure of pragmatism is required to get to the final configuration with a minimum of unnecessary complications.

As a guideline, a number of worked examples are set out in the associated repositories, which include adding tidal processes to configurations where the parent models did not have tides. For these simulations proper representation of tides was considered fundamental. For this we choose to use terrain following coordinates, to better represent shallow water processes, and tidal forcing from FES2014 (Lyard et al., 2021). For the South Asian, *SANH*[34] and South East Asian *SEAsia*[35] examples, this represents a significant departure from the parent model (without tides and computed on geopotential levels). Though many

---

[34]SANH: doi:10.5281/zenodo.6423211
[35]SEAsia: doi:10.5281/zenodo.6483231



other aspects of the parent and child configuration are similar. Another example, *SEVERN-SWOT*[36], details the workflow for (one-way) nesting of a 500m resolution child configuration in a macrotidal coastal regime.

## 3.2 Obtain code, compile model executables and build tools

This step is to obtain the code and build the required NEMO supported tools. This process is generic for all configurations. This step is largely software maintenance, rather than natural science, and can be largely automated once a successful workflow has been established. Official NEMO guidance can be found on the product webpages[37]. As a necessary precurser to subsequent steps, we provide a complimentary worked example, for a simple physics-only build on ARCHER2 (archer2.ac.uk) with NEMO4 that can be built with the linked scripts[38] and associated wiki documentation[39]. Static snapshots for these weblinks exist in *SEAsia*[40]. From experience we note there is no single 'best way' to structure the directory tree and flexibility should be encouraged according to the simulation or simulations required.

At this point it is worth briefly mentioning how users can implement choices in modelling frameworks, such as NEMO, since some choices are made at compilation and are therefore hardwired into the executable. In NEMO there are two tiers of hardwired choices. At the upper tier choices are made that activate NEMO modules, in addition to the core ocean (OCE) physics module, for example enabling ice or biogeochemistry capability. (See Sec. 4). These modules are then compiled together. At the lower tier, choices are made *w*ithin modules about which code blocks need to be compiled. These are set with compiler keys[41] and are used to activate, for example, MPI capability (`key_mpp_mpi`) and XIOS coupling (`key_iomput`). However, the majority of parameters that the user will edit, typically those which define details of and control the simulation (e.g. timestep and duration, forcing data location, parameterisation choices and coefficients) are contained within namelist files that are read at run time. Therefore for many applications, on completion of these compilation steps, the resulting NEMO executables and tools can be used for many configurations. The general direction of travel for NEMO is away from compiler keys to run-time configuration. Further NEMO specific details are given in the worked examples.

## 3.3 Model domain and geometry

After planning the new configuration and successfully building the machinery needed to run it, the following stage is to identify the most appropriate model domain, horizontal and vertical resolutions, and discretization schemes needed to adequately resolve the spatial scales and the processes the new configuration is targeting. The final outcome of this crucial stage is the definition of the 3D geometry of the model domain, including the horizontal and vertical coordinates and the 3D grid spacings, which typically vary for variables defined on the different faces, corners and centre points of the grid cells.

---

[36]SEVERN-SWOT: doi:10.5281/zenodo.6469990

[37]Official NEMO guidance: Setting up a new configuration (accessed 27 May 2022)

[38]https://github.com/NOC-MSM/SEAsia/tree/v0.1.0/SCRIPTS (accessed 27 May 2022)

[39]https://github.com/NOC-MSM/SEAsia/wiki/1.-Setting-up (accessed 27 May 2022)

[40]SEAsia: doi:10.5281/zenodo.6483231

[41]example compiler keys (accessed 27 May 2022)





Starting from version 4.0, NEMO loads at runtime an externally generated domain configuration file containing all the relevant grid geometry information. This separation of the grid generation process from the dynamical core permits a flexible approach to grid construction. If the planned configuration is based on an existing NEMO configuration (or idealised geometries), then the work to build a new domain configuration file can be done by tools and guidance that are supplied with NEMO. However, if the proposed work is for a new regional configuration, as is the main theme of these workflows, then the guidance outlined below is indispensable for directing the process.

- Location of boundaries: For the worked example, these are chosen with consideration of the tidal harmonics. It was verified (using TPXO9 (Egbert and Erofeeva, 2002), FES2014 (Lyard et al., 2021) or other tidal product) that none of the 4 largest semi-diurnal or 4 largest diurnal species had amphidromes near the proposed boundary, since for fixed relative errors in tidal amplitudes, small absolute errors at the boundary could scale to large absolute errors in the interior.

A principle of regional ocean modelling is to nest with the parent model in the deep ocean. Ocean-shelf exchange processes are complex, fine-scale and exert a first-order control on shelf seas (Huthnance, 1995), so it is expected that they would be better represented by the child than parent model. This choice comes with two penalties. The deeper water necessitates a shorter barotropic time step and steep continental slopes can cause issues of horizontal pressure gradient error for terrain following coordinate models. So the alternative of nesting on-shelf is also preferred for some studies (e.g. Holt and Proctor, 2008).

- Boundary bathymetry: In a regional model, the boundary interface with the parent model is a likely source of instability, especially if the grids are different. In light of this, the boundaries can be chosen to avoid grid-scale highly irregular bathymetry near the boundary (small islands or sea mounts and steep bathymetry) and to seek near orthogonal intersections between the boundary and the features, as done in the SEAsia configuration. An alternative is to precisely match the bathymetries at the boundary, with the child being interpolated to the target resolution, and then having a halo region inside the boundary where the bathymetry is smoothly transitioned to the target child bathymetry. However this sophistication is not required for our examples.

- Bathymetry preprocessing: The final bathymetry will be mapped onto your target grid. For a wide area domain this may require patching together bathymetries from separate surveys or, more likely, using a gridded product where merging has already been performed. It is worth checking the data for spatial discontinuities and applying appropriate filtering (though being cautious not to oversmooth lengthscales explicitly resolved at the model's resolution). Conversely bathymetry on the target resolution may adversely affect the large scale flow (e.g. restricted flow through narrow straits) or produce instabilities (model levels get too thin with ebbing tide), or generate spurious currents for terrain following coordinates over steep bathymetry, and so user defined modifications to the bathymetry might be necessary, according to the target grid. Note that the absence of systematic recording of the steps taken for bathymetric preprocessing is an endemic problem in model reproducibility that should be avoided.





- Bathymetry reference Datum: An important aspect of processing bathymetry data, for use in a numerical model, depends
        on the vertical vertical reference datum in the source grid. Configurations with large tides and/or surges need to pay
        particular attention to the datums of the source bathymetry. Often the bathymetry is referenced to Lowest Astronomical
        Tide (LAT), e.g. EMODnet (Lear et al., 2020). While it is useful for chart data to be referenced to LAT for navigational
        purposes, the bathymetry needs to be referenced to Mean Sea Level (MSL) for modelling purposes. There are however
fundamental limitations on the accuracy with this process since LAT is not accurately known in the absence of multiyear
        tidal records. Therefore the process of referencing and de-referencing the bathymetry to LAT is problematic. Nevertheless
        it may be achieved to first order from a long multi-decade integration of a tide-only model of the region of interest by
        using the lowest obtained sea level as a proxy for LAT.

        - Negative bathymetry and reference geopotential height: For configurations that involve inter-tidal zones, the bathymetry
can be negative relative to MSL. To deal with negative bathymetry NEMO can use a reference geopotential level defined
        at some height above MSL, so that all potentially wet points are below this reference level. Care is required when
        generating initial conditions and stretching of vertical coordinates to take into account the use of a non zero reference
        depth.

        - Critical depth for wetting and drying: In NEMO there is the option to allow for a grid cell to dry out as the tide ebbs.
This is implemented in practice by limiting the fluid flux out of the cell when a user defined minimum depth is reached.
        The specification of this minimum depth will be application dependent (typically a few cm's) and requires a compromise
        between maintaining numerical stability, for a given time-step, against enhanced realism of thinner critical depths.

        - Grid discretisation: When designing the computational mesh, the lateral extent of the domain and the 3D resolution are
        likely to be determined by the experiment requirements and the available HPC resources. In the horizontal direction,
NEMO supports structured quasi-orthogonal curvilinear quadrilateral Arakawa C-grids. These types of grids can offer a
        good degree of compromise between flexibility and accuracy, allowing one to improve the representation of many coastal
        processes (e.g. the propagation of Kelvin waves and land-ocean interactions through the aligning of grid lines with the
        coast (Adcroft and Marshall, 1998; Greenberg et al., 2007; Griffiths, 2013)). In addition, they can be used to refine the
        resolution in a specific location of the domain, to improve for example shelf-open ocean exchanges (e.g. Bruciaferri
et al., 2020). However, since they rely upon analytical coordinate transformations, they typically have limited multiscale
        capability in comparison to more versatile (e.g. triangular mesh) unstructured grids (Danilov, 2013; Holt et al., 2017). In
        the vertical direction, NEMO takes advantage of quasi-Eulerian generalised vertical coordinates $s(x, y, z, t)$, where the
        time dependence allows model levels to "breath" with the barotropic motion of the ocean. In domains with shallow seas
        and tidal dynamics, a species of terrain following coordinates are often adopted in order to provide vertical resolution
to resolve highly dynamic tidal processes on the shelf (Fig.2) as well as being able to resolve the open ocean forcing
        conditions and structured water mass properties (e.g. see Wise et al., 2022). In a linked example (e.g. *SEAsia*[42]) we chose

---

[42]SEAsia: doi:10.5281/zenodo.6483231

75 levels of hybrid z-sigma vertical coordinates. These were configured so that below the 39th level (at around 400m) the coordinates would transition to z-partial step so as to favourably compare with the parent z-partial step.

- Process oriented experiments: It is often useful to conduct simple numerical experiments to assess whether the chosen 3D model geometry is numerically stable and accurate enough for the target application. For example, steeply sloping model levels can introduce errors in the computation of the horizontal pressure force (e.g. Mellor et al., 1998). In such a case, conducting idealised horizontal pressure gradient tests can be instructive to ensure that the chosen vertical discretization scheme does not introduce undesirable spurious velocities (e.g., see experiment 1 of Bruciaferri et al. (2018) or Wise et al. (2022) for the details in idealised or realistic scenarios, respectively). Similarly, geopotential z-coordinates can introduce excessive spurious entrainment and mixing when simulating gravity currents (e.g. Legg et al., 2006). Therefore, idealised cascading experiments similar to the one of Wobus et al. (2013) or Bruciaferri et al. (2018) can be useful to reveal excessive dilution of dense overflows. Finally, tide-only forced experiments in barotropic (first) and stratified (after) set-ups can be extremely useful to early detect issues in the model geometry that could negatively affect the accuracy of the simulated tidal dynamics (see experiment 3 of Wise et al. (2022) for details).

Worked examples are given for the *SANH*[43], *SEAsia*[44] and *SEVERN-SWOT*[45] domains. For these new configurations, initial tests were conducted to ensure that horizontal pressure gradient errors were acceptable and that the tides were simulated accurately. Having addressed any emergent problems with these tests, additional complexity can be sequentially added: realistic initial conditions; realistic temperature, salinity and velocity open boundaries; meteorological forcing and finally freshwater forcing.

In summary, the steps to create the domain file are to first create a set of coordinates for the target grid, then make a bathymetry for these coordinates. Finally extend the domain in z-direction, with the chosen type of vertical coordinates, to complete the 3D discretisation of the domain.

Note the domain configuration file is static with respect to time. Any time variability in, for example, the vertical grid can be captured at run time with the output files.

### 3.4 Initial conditions

Initial conditions can be idealised or realistic. Effective use of appropriate initial conditions can expedite the spin-up of a model in slowly evolving regions of the domain (e.g. deep water salinity). However the initial conditions are constructed, it is likely that they are imperfect and that at least some spin-up time is required for dynamical adjustment on the child grid. For example the default initial condition machinery in NEMO uses only temperature and salinity with the expectation that the velocity field can be spun up from rest.

An alternative is to pose a "soft-restart" state rather than initialising the model from rest. In the "soft-restart" method, salinity, temperature, current velocities and sea surface height from, for example, a coarser resolution model or a reanalysis product

---

[43]SANH: doi:10.5281/zenodo.6423211
[44]SEAsia: doi:10.5281/zenodo.6483231
[45]SEVERN-SWOT: doi:10.5281/zenodo.6469990





are interpolated into the model grid and are then used to create a pseudo-restart file. Using this method requires a shorter spin up to allow the model to adjust to any instabilities. The worked example in the *SEAsia* repository details how to generate a
pseudo-restart for a reanalysis product.

A principle is to match initial condition and lateral boundary condition data (for temperature and salinity). A mismatch here can generate density driven currents tangential to the open boundary, which may persist long into the simulation under geostrophy. The most common potential challenges can be grouped into issues arising from inconsistencies across products:

- Parent to child grid interpolation: The parent and child grids are likely to be different. Therefore the coastline and bathy-
metric features will likely have different representations such that the child might have land points where the parent has wet points, or vice versa. Flood filling prior to interpolation (lateral filling of all land points with nearest neighbour wet point values) and downfilling isolated canyons (using e.g. SOSIE[46] tools) can address issues of bathymetric representation following interpolation. Additional smoothing of tracer fields may also be required if, for example, new straits are opened by the child grid. Furthermore, the representation of the ocean interior might be different between the two grids.
For example, a pycnocline might be poorly represented between two thick levels in the parent grid, but how should this "step" be represented in a finer resolution child with increased vertical resolution? Whatever method scheme is chosen, it is likely therefore that some spin-up will be required to let fine-scale features evolve. This spin-up should be of similar order to the flushing time of the shelf sea basin (often a few years).

- Equation of state: With a non-linear equation of state, interpolating temperature and salinity onto a child grid will not
ensure preservation of static stability. Alternatively the equation of state that generated the parent data might be subtly different from the equation of state in the child model. Though these effects are likely to be small and quickly dissipate, in practice they have been seen to trigger convection in marginally stable environments. So checking for static stability of the initial condition is recommended if stability issues arise in the first few timesteps.

However the initial conditions are constructed, it is likely that they are imperfect and that at least some spin-up time is
required for dynamical adjustment to the child grid to occur.

Even if the target initial conditions are prescribed as being from a "realistic" source, it can be an instructive and time-saving route to a final configuration to start with idealised initial conditions. NEMO has the facility to compile user-defined initial conditions into the executable which can be invoked by namelist parameter choices at run time. In the supporting *SEAsia* repository, two examples for idealised initial conditions are used: a) domain constant temperature and salinity; b) horizontally
homogeneous temperature and salinity, constructed from the World Ocean Circulation Experiment (WOCE) climatology to be broadly representative of the region. The latter is used to assess the horizontal pressure gradient errors in an unforced run, thereby testing the limitations of the vertical discretisation.

---

[46]SOSIE tools (accessed 27 May 2022)





## 3.5 Open boundary conditions

The lateral boundaries are the points that define the horizontal extent of the model domain. Information must be specified at
these points to constrain the interior solutions, effectively providing a forcing to the model. When the regional model differs
from the model that was used to generate the boundary data, which typically is the case, differences between the interior
solutions will emerge. An open boundary is a way to specify the external forcing while allowing phenomena produced within
the interior domain to exit across the boundary without disturbance. In some sense open boundaries allow the physical domain
to extend beyond the boundary of the computational domain, for example by allowing a wave to exit the domain without
reflecting back into the domain.

It is important to recognise that the formulation of open boundary conditions tends to be based on simplified physics,
focusing in particular on the hyperbolic part of the dynamics. In general these open boundary conditions will not be perfect and
care must be taken to assess instabilities and model accuracy deficiencies attributable to the boundary conditions. For example,
a parent model that is eddy rich may result in data that appears noisy, leading to a mismatch in dynamics at the boundary.

NEMO offers a number of namelist options to specify different open boundary condition formulations as well as set the
frequency of the supplied data. This data typically comes from an external parent model with a much lower frequency (typically
daily, 5-daily or even monthly for global products). There is an option to interpolate in time.

It is possible to specify 'structured' open boundaries that define the Northern, Southern, Eastern and Western boundaries, as
well as 'unstructured' open boundaries. While the former is useful in idealised setups, unstructured boundaries enable complex
geometries defined by a supplied coordinates file. In cases where different boundaries have different requirements, it is possible
to define multiple sets of unstructured open boundaries that can use different namelist options and datasets.

The namelist is organised so that boundary conditions are separated into the 2d depth mean velocities and sea-surface height,
the 3d depth dependent velocities (perturbations from the depth mean) and the 3d tracer fields. Tidal harmonics can also be
specified as part of the 2d fields. Following the principle of building up complexity, it is worth configuring the open boundaries
for the depth mean velocities first. This can include tidal harmonic forcing.

The choice of boundary condition, for the 2d velocities, is primarily a choice over which radiation condition to use (e.g.
Flather (1976), or an Orlanski (e.g. Marchesiello et al., 2001) scheme). For the 3d velocities and tracers, one can also choose
to relax the child field to the external data over a buffer zone or apply a condition on the normal flux or normal gradient. For
example it is possible to apply a radiation condition to the 2d velocities, flow relaxation scheme to the tracers and zero gradient
to the 3d velocities.

The key considerations are whether the open boundaries are affecting either stability or accuracy. Some specifics to consider
include:

- Parent-to-child: The boundary data will likely be associated with bathymetry that is different from that used by the child
  model. This can result in differences between the parent and child in terms of transport across the boundary. It may
therefore be beneficial to match the bathymetry along the boundary. Another consideration is whether to preprocess the





velocities so that the transports in the child match those of the parent when regridded. Note that NEMO allows the user to provide files that contain the full velocities (2D + 3D), which it will then separate at run time.

The boundary data will also likely be associated with a vertical grid that is different from the child vertical grid. If they have not been regridded then NEMO provides an option to vertically interpolate onto the native grid at run time. For 3D velocities this could lead to inconsistency with interpolated tracer fields.

Changes in grid and bathymetry may also result in sections of boundary that are separated from land point by thin strips of wet grid points. This may result in spurious currents and a need to mask out certain grid points.

The boundary velocities may also need to be rotated. If the external velocities are specified as rectangular, for example, they might require rotation to be correctly oriented on a spherical grid.

There may also be temporal differences between the child and parent. Specifically, models can be set up to ignore leap years, which may result in the boundary data becoming out of synchronisation with the child model time.

Finally, even if the vertical grids are the same, mismatches can occur if different types of free surface are applied: Many regional applications use non-linear free surfaces whereas global models often use fixed z-levels. These effects are strongest in the surface layers and could be mitigated by constructing boundary conditions from volume fluxes, if appropriate.

- Tides: As previously noted tidal amphidromes should ideally be away from the boundary. Additionally, as previously noted, a mismatch in parent-child bathymetry can result in a mismatch in transports, this also affects transports due to tides. Relatedly, it should be ensured that tides are not present in the external boundary data if tides are also specified with harmonics.

- Volume conservation: Open boundaries can allow gain and loss of water through the boundaries which may result in drift in mean sea level, as well as accumulating dynamical errors. NEMO provides an option to maintain constant volume via a correction. For a model including tides, however, this could be considered inappropriate.

- Spurious currents: Spurious currents can be generated at open boundaries that appear trapped but may affect the interior momentum over time. Areas where the boundary intersects the continental margin are particular areas of concern because the sloping bathymetry can act as a wave guide for spurious variability. A further consideration is the effect on non-physical aspects of the model, such a biogeochemistry (see Section 4.1). High vertical velocities at the boundary may not be apparent due to flow relaxation at the boundary. However, tracers that are not relaxed at the boundary will feel the effect of spurious vertical currents.

Following the above guidance to build sequentially, whereby complexity is incrementally introduced, it can be instructive to include open boundary conditions with a sequence of developments. Our workflows lean heavily on the PyNEMO[47] community python tool. A tide only example (forcing by FES2014 tides, temperature and salinity are set constant, velocities are

---

[47]PyNEMO:https://github.com/NOC-MSM/PyNEMO [last accessed 4July2022]





initialised as zero and boundaries set to initial values) can be found in the *SEAsia*[48] and *SEVERN-SWOT*[49] repositories. The documentation includes generating the boundary conditions and running the model. For boundary conditions including 2D and 3D velocities as well as tracers see the *SEVERN-SWOT* repository. The documentation includes generating the boundary
conditions and setting the namelist.

### 3.6  Atmospheric forcing

In this discussion we consider atmospheric forced ocean models and atmosphere-ocean coupled models.

### 3.6.1  One-way atmospheric coupling

In the one-way (forced ocean) setup, it can be helpful to consider that the meteorological processes can either affect the
thermohaline properties (via heat and radiation fluxes, or precipitation and can be applied as boundary conditions in the tracer equations), or they can affect the ocean momentum (via pressure, wind speed). In uncoupled configurations, atmospheric boundary layer processes must be parameterised in order to compute fluxes with which the ocean can be forced. NEMO has a number of parameterisation options (or Bulk formulations): i) NCAR (Large and Yeager, 2004), designed for the NCAR forcing, but also appropriate for the DRAKKAR Forcing Set (DFS) (Brodeau et al., 2010); ii) COARE3.0 (see Fairall et al.,
2003); iii) COARE3.6 (see Edson et al., 2013); iv) ECMWF, appropriate for ERA5 data (Beljaars, 1995); v) ANDREAS (see Andreas et al., 2015). Alternatively if all the atmospheric fluxes are known, these can be supplied directly as surface boundary conditions.

In addition to choosing the appropriate source and type of surface boundary conditions, there are a few additional considerations to be borne in mind when preparing the atmospheric forcing data set for a target child model that has different spatial or
temporal discretisation from the parent: 1) Calendar stretching (e.g. 3 hourly forcing on a 360-day calendar being mapped to a gregorian calendar); 2) Land-Sea masks can differ between the parent atmospheric grid and child ocean grid. This is especially problematic when a coarse parent grid is naively interpolated onto a finer child grid. The mismatch in coastlines results in mis-representation of near coast heat fluxes and sea breeze dynamics, but can be alleviated using flood-filling techniques whereby extrapolation and interpolation is applied separately to land points and sea points. Though problems can still arise if the child
grid has islands that simply do not exist in the parent grid. Finally, subtle differences in the atmospheric data expected by the bulk formulae is a common source of error (e.g. reference levels, specific versus relative humidity, net versus downward long and shortwave radiation etc.). So, as is often the case when re-purposing a complex system by modification: "trust but verify".

### 3.6.2  Two-way atmospheric coupling

In regional two-way coupled atmosphere-ocean models, the information transferred to the ocean from the atmosphere is essen-
tially very similar to that provided in a one-way forced set up, when the fluxes are known. The solar and non-solar surface heat flux, mean sea level pressure and freshwater flux are transferred, as well as the momentum fluxes from atmosphere to ocean

---

[48]SEAsia: doi:10.5281/zenodo.6483231
[49]SEVERN-SWOT: doi:10.5281/zenodo.6469990



(Lewis et al., 2019a). Then either the surface temperature or surface currents, or both can be sent back to the atmosphere. The variables are exchanged between both models via a coupler such as OASIS3-MCT (Valcke et al., 2015), and interpolated between the two grids, typically using first-order conservative interpolation for scalars and bilinear interpolation for vector
fields.

The coupling frequency can be optimised by considering the region over which the model is being run and the features and dynamics that dominate that area. However it must be set to a value larger than the model timestep. Wang et al. (2015) use a 3 hourly coupling frequency for their climate atmosphere-ocean model located over the Baltic and North sea but Zhao and Nasuno (2020), found that a coupling frequency of hourly or sub-hourly better reproduced the SST and consequently stronger
convection during the passage of the MJO. In the regional coupled suite RCS-IND1 (Castillo et al., 2022), an hourly coupling frequency was used to capture the temperature diurnal cycle, however options to move to using a 10 minute coupling frequency are mentioned, as this might prove beneficial for modelling rapidly changing conditions as in squalls and tropical cyclones.

There is a risk that coupled atmosphere-ocean models may become unstable and drift when run over long periods of time due to the feedbacks between both models. To constrain the drifts, nudging or weakly coupled assimilation may be required.
However not all models require corrections, the decadal-scale run carried out by Wang et al. (2015) and 100-year simulation in Primo et al. (2019) being examples of this. Alternatively, mixed two-way and one-way forcing approaches can be applied if either coupling or direct forcing is not appropriate for the entire domain.

A worked example is given in the *SEVERN-SWOT*[50] repository using ERA5[51] data. The documentation includes these global data being cut down and manipulated for use in forcing a regional configuration and then running the model with one-way
meteorological forcing.

### 3.7   Terrestrial river forcing

This aspect of configuring a regional model is uniquely challenging: river flow data typically comes from gauges, which are typically far upstream from the model's coastal grid point, or from hydrological models, where the data are gridded but not necessarily at fine enough resolution for the target application (e.g. many global products have a 1/4 deg resolution). Prepro-
cessing freshwater data can be particularly time consuming so it is worth giving careful consideration to design choice options at the outset. Where possible, consistent atmospheric precipitation and riverine data are preferred for consistent freshwater budgets. For example, the JRA-55 (Tsujino et al., 2018) and the COREv2 (Large and Yeager, 2009) datasets have an accompanying freshwater river dataset. Furthermore, if riverine biogeochemical fluxes are required, these would be best coming from the same source that the precipitation and riverine freshwater fluxes, or at least being constructed in a consistent fashion (see
section 4.1.3 for specific guidance.

Having identified the data sources, and the corresponding model grid points for freshwater inputs (itself a heavily labour-intensive exercise that defies straightforward automation), there are further choices to be made regarding the implementation: i) How is the freshwater distributed horizontally - if the coastal outflow is a river delta, the freshwater load should be distributed

---

[50]SEVERN-SWOT: doi:10.5281/zenodo.6469990
[51]ERA5 data access (accessed 27 May 2022)





between the tributary channels. Similarly, if the volume flux is large and baroclinic dispersal processes are not resolved then unrealistic freshwater lenses can accumulate at the coastline. This can also be remedied by redistributing the freshwater flux across neighbouring grid cells. ii) How is the freshwater distributed vertically - the freshwater can be vertically mixed to a specified depth, or enter as a surface plume. (In all our high latitude and tropical applications we mixed the freshwater over the top 10m.) Increasing vertical diffusion at river points can be used to compensate for unresolved estuarine mixing. iii) What is the salinity and temperature of the river water: the default implementation is often to add freshwater (zero salinity) at the temperature of either the sea surface or the atmosphere, however, better results are likely if observed or river model values are used. These choices can be adjusted to achieve target temperature and salinity characteristics of the plume. To date there is no accommodation for ground water fluxes, though these can be considerable (Zektser and Loaiciga, 1993). Finally, validation to ensure accurate estuary-mouth forcing is challenging. Satellite salinity has a resolution of approximately 35-50 km for the Soil Moisture and Ocean Salinity (SMO) and 100–150 km for the Aquarius satellite, whereas insitu measurements capture plume features (and freshwater intensifications) at scales O(10m) that are much finer that the resolved scale but lack spatial coverage. Where possible use should be made of near coastal buoy and survey data, but in the absence of this we must settle with far field validation against hydrography, accepting that this conflates the effects of circulation and surface forcing.

The worked example in the *SEAsia* repository details how rivers fluxes were taken from the JRA-55 dataset (Tsujino et al., 2018) and mapped to the nearest coastal grid point. Subtleties for large rivers include i) avoiding placement of domain boundaries near large river outflows, and ii) laterally spreading the coastal source points to represent deltas and also to avoid unrealistic numerical issues if outflow values are locally too large.

### 3.8 High Performance Computing: decomposition and optimisation

Modelling frameworks like NEMO are equipped to be run on High Performance Computing machines. This is facilitated by the optional abstraction of input-output procedures (XIOS) from the dynamical model (NEMO). These can then be separately optimised for the target machine and, crucially, most I/O to disk can be overlapped with the continuing computational tasks. Most HPC platforms will have multiple nodes each comprising of a number of CPU cores and some shared memory. However NEMO and XIOS have different computation and memory requirements. Making the best use of each HPC platform can be further complicated by the service's charging algorithm and possible non-linearities in performance scaling arising from subtleties in hardware design. Nevertheless, when configuring a simulation, there are only two sets of fundamental considerations:

1) How many compute cores will be allocated to the dynamical core and how many cores to assign to each node? Each core will have a roughly equal grid cell fraction of the surface map of the whole domain (parallelised subdomains contain the full water column).

2) How many cores should be dedicated to the XIOS processes and how many cores to assign to each node?

Some general guidance can be offered to address these questions:

- Computational resource splitting: The best ratio of XIOS to NEMO cores depends on the volume of data to be written. Since this is configurable at run-time via the XML files, some care is needed to provide the XIOS processes with

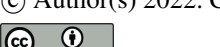



sufficient resources to cope with any expected variations. Though there are no easy answers as to how to optimise domain decomposition, a starting point would be to allocate cores between NEMO and XIOS in a ratio of $n : \sqrt{n}$. Often, fewer XIOS cores are required but each XIOS process is likely to require access to more memory than each ocean core. Typically, this is achieved by running XIOS cores sparsely populated on dedicated XIOS nodes or by using options to control placement of processes to CPUs on nodes running a mix of XIOS and ocean cores. The best solution will be hardware-specific. For example, on a system with 4 NUMA regions per node, you may choose to place one XIOS process alone in the first NUMA region (thereby giving it access to the maximum amount of shared memory) and to fully-populate the remaining NUMAs with ocean processes. Performance optimisation of the NEMO to XIOS decomposition comes with experimentation. Timing information provided in the `timing.output` file (generated when `ln_timing = .true.`), can be used to decide on the number of ocean cores by checking the average time per timestep figure for a variety of decompositions. Optimising the number of XIOS cores is usually more a case of finding the minimum number that will cope with the demands determined by the XML files. Insufficient XIOS resources may simply mean a longer than necessary wait at the end of a run but it can also lead to intermittent 'Out Of Memory' failures when disk I/O fails to keep up with the requests entering the queue. XIOS will report performance statistics at the end of a successful run.

- Domain decomposition: It is good practice to allow NEMO to automatically define the domain decomposition. This happens when `np_jpni` and `np_jpnj` are $< 1$. The model will take the number of ocean cores provided to the parallel run command (e.g. mpirun) and will minimise the horizontal subdomain size and maximize the number of eliminated land-only subdomains. It is likely that your first guess at the ideal number of ocean domains is not optimal; pay attention to messages in `ocean.output` for advice on better choices. See the section 7.3 of the NEMOv4.2 reference manual for more details. Experiment with a range of decompositions and use considerations of cost and "time to solution" to decide your best choice. As a rule of thumb, it is likely that performance will not scale with domains smaller than around $7 \times 7$ points since communication will begin to dominate over computation with domains smaller than this. However, this is only a guide and NEMO developments towards exascale computing are continuing to challenge these limits. In older versions of NEMO land suppression (to avoid computation over landpoints) had to be explicitly activated. This efficiency saving is automatically activated since NEMOv4.2 but is discussed in the worked examples (e.g. *SEVERN-SWOT*[52] unforced run example).

## 3.9 Troubleshooting

Invariably, during the course of developing a new configuration, a number of problems will be encountered where the model blows up or otherwise does not behave as expected. The general principle to more rapidly building a new configuration is to make changes slowly. In that, we mean, starting from something that works and sequentially testing incremental changes, rather than making multiple changes at once.

The most common problems new users encounter fall into two main categories: 1) compilation and start-up difficulties, and 2) run-time stability issues. Or, equivalently: 'getting past the first time-step' and 'getting beyond the first few inertial

---

[52]SEVERN-SWOT: doi:10.5281/zenodo.6469990





periods'. Often, difficulties in the first category arise from inconsistencies in the build environment. It is essential, for example, to compile XIOS and NEMO executables in identical machine environments and this must include netCDF4 libraries which have been compiled with full parallel support if all the functionality of XIOS is to be used. This same environment must then be used at run-time. It is always a good idea to test the environment and batch system with one or more of the NEMO reference configurations before tackling your own, bespoke, configurations. This will separate system issues from any issues introduced

by your own changes. One way to do this is to run all or parts of the SETTE testing suite. There is a full section in the NEMO user guide[53] explaining how to do this. At a minimum, it is recommended to run the GYRE_PISCES configuration which requires no external data files.

After this, any continuing start-up issues with bespoke configurations should be isolated to bespoke code changes, errors in the inputs files (including namelists) or resource size issues (e.g. your model requires more memory than is available on

the nodes you are assigning). Precise advice will be machine-specific and depends on how the architecture is configured, but both XIOS and NEMO will detect and report many issues. Check both the batch output logs and the ocean.output file(s) for messages. Use the Discourse[54] message channels for community support in interpreting any failure whose cause is unclear. If failures are sudden, unlocated in the code and catastrophic, then recompile NEMO with compiler debug flags and try to obtain a traceback, which identifies where in the code the crash occurs.

Once past the first time-step, the next failures are usually due to numerical stability issues or I/O issues at a future checkpoint. Even the best prepared initial conditions are likely to trigger fast-moving responses (e.g. coastal trapped Kelvin waves) at start-up. Often, any numerical stability issues associated with this initial adjustment can be avoided by running a short period with a reduced timestep. A more general section on model stability constraints is included below. I/O issues at a future checkpoint will either be errors in the input files (e.g. the model reaches a point when it requires the next month's surface forcing, but

those files are not available) or output issues associated with specific output. Examples of the latter include XIOS/netCDF4 detecting 'not representable' values (usually a good indication that NaNs are occurring in the simulation) or memory limits being reached due to a sudden surge in output requests. If the latter occurs in an established simulation with no new demands on XIOS, then it may be indicative of external influences changing the usual rate of I/O to disk (faulty switches etc.). Here we touch on some general issues that can arise.

**Syntax errors in XIOS**: The XML files which control the XIOS output are very sensitive and susceptible to syntax errors. Commonly, syntax errors in the XML files manifest in a lack of output, or a model failure when writing output. To avoid these, difficult to trace, issues, it is strongly recommended that you edit the XML files independently of other changes being introduced to a working configuration and test each change in short test simulations.

**Model stability constraints**: As with most explicitly formulated computational fluid dynamics codes it is critical that the

timestep is shorter than the time it takes for substance to cross a grid box, so that the flow of mass, tracer, or the propagation of waves are resolved. In configurations with deep water and a free surface, the shallow water wave speed will likely be the

---

[53]NEMO user guide (SETTE): https://sites.nemo-ocean.io/user-guide/sette.html

[54]NEMO Discourse: https://nemo-ocean.discourse.group [last accessed 21 Jul 2022]





timestep limiting process. Accordingly, the timestep will linearly decrease with refinement to the horizontal resolution. This can make a fine-scale coastal model with some deepwater grid-points computationally expensive.

Similarly, horizontal diffusion operators limit the time step. Though here, it is the square of the grid spacing ($\Delta x$) that limits the timestep ($\Delta t$). For example, a Laplacian diffusion operator would scale $\Delta t < \Delta x^2 / K_{lap}$. So a factor of 2 reduction in grid spacing would correspond to a factor of 4 reduction in timestep. This can be offset by a linear reduction in diffusion coefficient $K_{lap}$ (units $m^2/s$ and represented as a velocity scale multiplied by a length scale). Similar analysis can be performed for bilaplacian diffusion (with coefficient $K_{bi}$, units $m^4/s$ such that $\Delta t < \Delta x^4 / |K_{bi}|$. There are many other more subtle and nuanced stability constraints relating to most areas of the dynamics, understanding these generally requires detailed numerical analysis, so are usually left to trial and error. Nevertheless the process can be facilitated by inspection of CFL diagnostics (for example, activated through the namelist parameter `ln_diacfl`).

**Code errors**: Despite best endeavours, the code base can have errors (especially, for example, on branches that are being actively developed). Also, user implementation errors (e.g. inappropriate coefficients, as discussed above) can be hard to distinguish from parameterisation failures, which can both typically have explanations that are attributable to physical characteristics of the domain (e.g. the simulation blows up in a strait / over a sea mount / at a coastal inlet / on a spring tide / following intense stratification events). On the other hand, code errors arising from incorrect indices or variables may appear to have a more random behaviour (e.g. crash point moves with a changing number of cores, or otherwise "surprising" or non-Newtonian behaviour).

With any advanced technology, troubleshooting is an inevitable part of developing. Therefore document your working; revert to a last working setup (if it exists) when things do not go as expected. Finally, making sure your issues are reproducible will help if you seek assistance.

## 4    Additional modules

One of the strengths of the NEMO framework is the broad community user base that actively develops additional modules that augment and enhance the value of the physics-only ocean simulations. In the following, experiences are shared relating to biogeochemical, wave and ice modelling.

### 4.1    Biogeochemistry

A significant challenge of regional biogeochemical modelling is in initialising all the necessary fields and supplying appropriate surface and lateral boundary conditions, along with river inputs. This is complicated by the fact that data for initialising and forcing biogeochemistry models can come from a variety of sources and usually require some pre-processing before it can be applied. In particular, special attention should be paid to the data units, which might need to be converted according to the model-specific requirements. Specifically, concentrations should not be confused with fluxes.




A number of data sources are publicly available, both observational and modelled. Widely used observational global products include: World Ocean Atlas[55] (Boyer et al., 2018b), containing monthly gridded inorganic nutrients and dissolved oxygen in the upper 800m and annual values deeper; and GLODAP[56] (Olsen et al., 2016), containing annual gridded dissolved inorganic carbon and total alkalinity, as well as inorganic nutrients and dissolved oxygen. Modelled products, such as CMS global models, using PISCES, contain simulated nutrients and phytoplankton fields. For regionally specific products data may be available from regional consortia and programmes (e.g. North Sea Biogeochemical Climatology (NSBC[57]) (Hinrichs et al., 2017) and the EMODnet portal[58] (Lear et al., 2020)). However, observational sources include limited number of measured parameters, while modelled data is likely to be generated by a model with a different (usually simplified) trophic structure compared to the target biogeochemical model. Assumptions will therefore have to be made when generating initial and boundary conditions to infer the required fields based on the available data; application of the Redfield ratio can be a valuable tool to preserve the stoichiometry when deriving the relative elemental composition of living organisms. Ideally, physics and biogeochemistry boundary and initial conditions would come from the same source. When using data from difference sources, care must also be taken to avoid, or manage, potential mismatch issues between, for example, the depth of the nutricline and the pycnocline. In these circumstances one would expect the simulations to require a spin-up period. The length of spin-up will depend on various aspects of the region's dynamics and quality of the starting conditions of the key state variables. For example, a timescale of several months is typically sufficient for most pelagic variables, whilst certain benthic variables can take several years or even decades to reach equilibrium. As with physics-only regional models, attention needs to be applied to appropriately formulating the model forcings and inputs. Generalised considerations when initialising a biogeochemistry simulation are presented in the following subsections.

### 4.1.1 Initial Conditions

It is generally preferable to initialise the biogeochemistry in low biological productivity periods. This reduces model uncertainty as the phytoplankton and zooplankton are at their lowest values and nutrients are mainly present in their inorganic form. In the absence of appropriate data, distributions of phytoplankton functional groups can be estimated from measured chlorophyll. Other pelagic variables for which detailed data is not available (e.g. concentrations of dissolved organic carbon (DIC), calcite and bacteria) are typically initialised to constant values, ideally derived from any available observational data in the region. Distributions of benthic variables are often difficult to estimate as there is usually little data available. Climatological profiles can be used but may not be representative of the target region of interest. It can take years, even decades, for some components (such as organic matter and macrofauna) to adjust. Our experience suggests that underestimated organic matter content will increase quicker in shallower, productive areas than overestimated content takes to decrease in deeper regions.

[55]World Ocean Atlas: https://www.ncei.noaa.gov/products/world-ocean-atlas [last accessed 20 Jul 2022]
[56]GLODAP: https://www.glodap.info [last accessed 20 Jul 2022]
[57]NSBC: https://www.cen.uni-hamburg.de/en/icdc/data/ocean/nsbc.html [last accessed 20 Jul 2022]
[58]EMODnet: https://emodnet.ec.europa.eu/en [last accessed 20 Jul 2022]





### 4.1.2 Surface Boundary Conditions

The biogeochemical fields required for surface boundary forcing will depend on the region of interest and the target application. By way of example, nitrogen deposition is an important source of nutrients to the system and is commonly prescribed. Model products (e.g EMEP for Europe, HTAP for Global (Simpson et al., 2012; Tan et al., 2018)) are a useful data source. In shallow water regions light attenuation due to coloured dissolved organic matter (Gelbstoff absorption) can affect the radiation budget (e.g. Kara et al., 2005). It can be prescribed using data from ocean color observations such as those provided by OC-CCI (Sathyendranath et al., 2019). Atmospheric $pCO_2$ data should be provided for any model that includes a carbonate system. Observational data can be converted from the $fCO_2$ product available through SOCAT (Bakker et al., 2016). In some areas iron deposition may be important and can be included through modelled products such as provided by the GESAMP model ensemble (Myriokefalitakis et al., 2018). Though it is important to note that since only a small fraction of iron dust that lands on the ocean surface becomes bioavailable, any surface boundary values should be adjusted accordingly.

### 4.1.3 River Input

Observational data are (usually) the most precise source of data, but are limited in availability. In general, it is better to use the same source for all rivers included in the model setup. Therefore modelled products such as GlobalNEWS (Mayorga et al., 2010) and HYPE (Arheimer et al., 2020) can be the best option, though they will require, or implicitly make, some assumptions regarding land use. In some cases excessive anoxic conditions can form around river mouths, typically with larger rivers in regions already subject to anoxia. Therefore, if riverine oxygen is needed but is unavailable, impose a concentration around the saturation value. Any data on river inputs will usually contain a limited number of variables compared to requirements of the target model. Therefore, assumptions will have to be made, for example partitioning total nitrogen inputs into various inorganic and organic fractions. A worked example generating river forcing (discharge and nutrients) from the GlobalNEWS2 model is given in the supporting repositories *SANH*[59] and Partridge (2022b).

### 4.1.4 Lateral Boundary Conditions

Lateral boundary condition forcing data should be obtained similarly to the initial conditions data. Typically a flow relaxation scheme (FRS) is used as a boundary condition for fields where values are available, which usually include inorganic nutrients, DIC, total alkalinity and dissolved oxygen. Other variables are conserved within the domain by using a zero-flux (Neumann) condition. However, this can fuel spurious behaviour around the boundaries (e.g. non-depleting high levels of nutrients at the boundary can lead to phytoplankton blooms in the region nearby). This can be mitigated by setting low boundary values for phytoplankton. Long-living material, such as organic semi-labile and semi-refractory organic matter, can propagate into the domain over decadal simulations if applied at unrealistic levels. Particular care on the treatment of depth varying velocities at the boundaries is required when including biogeochemistry in the configuration. Unconstrained tracers that the boundary are

---

[59]SANH: doi:10.5281/zenodo.6423211



susceptible to instabilities from erroneous vertical velocities, which would not be seen in constrained tracers, and hence not apparent in the physics-only simulation with boundary constrained temperature and salinity.

### 4.1.5 Output testing

Before committing to production simulations a prudent initial test can confirm the simulation conserves mass of a tracer and 820 that the boundary conditions are correctly applied. To this end we suggest the following procedure:

Create two passive tracers (TRC1, TRC2) with a uniform initial value (e.g. 1.0). Using Neumann boundary conditions for TRC1 confirm the tracer is conserved. Using FRS boundary condition for TRC2, with a low constant boundary value (e.g. $1E - 10$), confirm the low values are not confined to the boundary after 10 time-steps.

When changing machine or domain decomposition it is advisable to check for bit-identical solution in test simulations. 825 Finally, be vigilant for negative or unrealistic values in areas that typically could cause problems. Such regions include boundaries, river mouths, or coastlines that might 'trap' and accumulate tracers.

### 4.1.6 Biogeochemical models used in regional configurations

Here we highlight just three biogeochemical models, with which we have experience and that span a range of complexities in biogeochemical process representation.

**ERSEM specific considerations**

The European Regional Seas Ecosystem Model (ERSEM[60] [last accessed 20 July 22])) (Butenschön et al., 2016) was originally designed as a marine biogeochemistry and the lower trophic levels model for the temperate European shelf seas (Baretta et al., 1995; Wakelin et al., 2020). However it has found application in global settings (de Mora et al., 2016) as well as in regional configurations in the other parts of the world ocean, using a variety of hydrodynamic models, e.g. in the Mediterranean (Kay 835 et al., 2020), Arabian (Sankar et al., 2018), and East China Seas (Ge et al., 2020).

ERSEM describes the biogeochemical cycling of carbon, major macronutrients (N, P, Si), and, optionally, iron. One of the distinct features of this model is that the carbon to nutrient ratios of functional groups are not fixed, varying depending on the environmental conditions. It also includes carbonate system and detailed description of microbial food web. Standard ERSEM configuration includes 52 pelagic tracers and 36 benthic variables. ERSEM's detailed benthic module is relatively unique to 840 biogeochemical models, however if the appropriate initial conditions are not available this can require an extended spinup time. Within ERSEM, the units of carbon are specified in mg, whilst other nutrients are specified in moles. Care should therefore be taken when applying, e.g. the Redfield ratio, to ensure state variables are presented in the correct units.

ERSEM adopts the modular and scalable structure of the Framework for Aquatic Biogeochemical Models (FABM[61] [last accessed 8 Jul 2022]). In NEMO the coupling happens within the `TOP_SRC` routines via FABM driver, with the necessary steps 845 and changes being documented in the wiki of the NEMO4.0-FABM repository (https://github.com/pmlmodelling/NEMO4.

---

[60]ERSEM: https://github.com/pmlmodelling/ersem
[61]https://github.com/fabm-model/fabm



0-FABM/wiki [last accessed 12 Jul 22]). Worked examples for setting up ERSEM for the SANH and SEAsia domains are given in the linked repositories[62,63].

**PISCES specific considerations**

PISCES (Pelagic Interaction Scheme for Carbon and Ecosystem Studies) is a biogeochemical model which simulates the lower
trophic levels of marine ecosystems (phytoplankton, microzooplankton and mesozooplankton) and the biogeochemical cycles of carbon and of the main nutrients (P, N, Fe, and Si) (Aumont et al., 2015). It is the default biogeochemistry module for NEMO, and is distributed with the code base. (It also ships with CROCO, Coastal and Regional Ocean Community model[64], which is based on ROMS AGRIF). PISCES is designed for regional and global applications. It has 24 tracers including two phytoplankton compartments (diatoms and nanophytoplankton), two zooplankton size classes (microzooplankton and mesozooplankton)
and a description of the carbonate chemistry. Optional parameterizations can be activated to control the complexity of iron chemistry or the description of particulate organic materials.

Examples where PISCES has been used to study specific regions include the Peru upwelling (Echevin et al., 2020), the Indian Ocean (Resplandy et al., 2012), and the Mediterranean (Richon et al., 2018). The PISCES community[65] offers training materials that is suitable for beginner and advanced users.

**MEDUSA specific considerations**

The Model of Ecosystem Dynamics nutrient Utilisation, Sequestration and Acidification (MEDUSA)[66] (Yool et al., 2011) was originally developed as an "intermediate complexity" marine biogeochemistry model for global-scale, high-resolution applications (e.g. Popova et al., 2010). Its design is intended to be intermediate between simple nutrient–phytoplankton–zooplankton–detritus (NPZD) models (e.g. LOBSTER; Lévy et al., 2005), and more sophisticated plankton functional type
(PFT) models (e.g. ERSEM; Butenschön et al., 2016). It represents the biogeochemical cycles of nitrogen, silicon, iron, carbon, alkalinity and oxygen (Yool et al., 2013). MEDUSA has 15 pelagic passive tracers, static 3D fields of carbonate chemistry and includes a simple "bucket scheme" for 2D benthic reservoirs. In NEMO the coupling happens within the TOP_SRC routines. For many applications, MEDUSA and PISCES are broadly similar, but are developed by different communities and thereby offer a level of model diversity.

Since MEDUSA is a reduced complexity biogeochemistry model all the variables for the lateral boundary conditions can be supplied by parent data. This means the boundary conditions can be simplified by specifying a relaxation condition on all (physics and biogeochemistry) variables. MEDUSA coupling with NEMO is effectively one-way so, for example, phytoplankton blooms will not affect the absorption of shortwave radiation and influence upper ocean temperatures.

---

[62]SANH-BGCsetup: https://github.com/dalepartridge/SANH_BGCsetup [last accessed 12Jul22]

[63]SEAsia-BGCsetup: https://github.com/dalepartridge/ACCORD_SEAsia_BGCsetup [last accessed 12Jul22]

[64]CROCO: https://www.croco-ocean.org [last accessed 22 Jul 2022]

[65]PISCES Community: https://www.pisces-community.org

[66]MEDUSA: https://code.metoffice.gov.uk/trac/medusa [last accessed 21 Jul 2022 - requires a Met-Office science repository account]





Though MEDUSA was conceived as an efficient biogeochemistry model for global applications in NEMO, it has also been
used in a regional configuration. A worked example for setting up NEMO-MEDUSA in a western Indian Ocean domain is
given in the linked repository[67].

## 4.2 Waves

Surface wave processes mediate the transfer of energy and momentum between the atmosphere and the ocean. They are
associated with surface intensified fluxes of momentum and energy that gives rise to a Stokes drift (Stokes, 1847), which makes
them an essential component when modelling particle drift. Waves can also modify subgridscale mixing (e.g. by injecting
turbulence into the surface layers, through waves breaking (Craig and Banner, 1994), or modify the character of the turbulence,
into Langmuir turbulence (Belcher et al., 2012)); they can affect the horizontal momentum in the Ekman layer, though the
interaction with planetary rotation, (Hasselmann, 1970; Polton et al., 2005). In near coastal scenarios waves can give up
their energy and momentum accelerating a mean flow or inducing "wave setup" (a change in background sea level) and can
exert forces on the bathymetry, known as "radiation stress" (Longuet-Higgins and Stewart, 1964). Studies demonstrate value in
coupling waves in regional operational systems. For example, in drift simulation (Bruciaferri et al., 2021), but also in modifying
surface temperatures by changing the depth to which the warm surface waters are mixed (Lewis et al., 2019b). Coupling also
impacts on the momentum transfer at the air-sea interface via alteration in surface roughness and stress, which affects the wave
growths (Valiente et al., 2021).

When adding waves to a regional ocean setup, a number of considerations need to be borne in mind when designing the
configuration for the target application.

Spectral wave models are relatively expensive to run. Though spatially only two-dimensional, the spectra have frequency
and directional discretisation (and a different timestep), which typically doubles the cost of the ocean-only component (e.g.
Lewis et al. (2019a) and Hashemi et al. (2015)).

This cost can be mitigated by the level of coupling used; if the waves and the ocean do not significantly affect each other
then the components can be run independently, or if adequate products exist, and the ocean component does not significantly
affect the waves, then data could be downloaded from e.g. CMS catalogue. Cost can also be mitigated in coupled ocean-waves
configurations by decreasing the rate at which variables are exchanged between modules.

Having determined whether or not the wave field needs to be simulated, the level of coupling can then be chosen. There are
three "zones" where coupling can be implemented:

1) At the air-sea interface:

    a. Waves act as a buffer to momentum exchange between the atmosphere and the ocean, for example reducing the surface
       stress experienced by the ocean when waves are growing. Typically this coupling is modelled one-way with a modifica-
       tion of water-side surface stress in the ocean module by wave growth and dissipation.

---

[67]NEMO-MEDUSA:https://github.com/NOC-MSM/Regional-NEMO-Medusa/ [last accessed 21Jul22]





b. The presence of waves facilitates the transfer of momentum into the ocean. This can be captured by a wave-height-dependent ocean surface roughness in the ocean module.

    c. Breaking waves inject turbulence into the upper ocean. This can be captured through modification of surface subgridscale turbulent kinetic energy in the ocean module.

    d. Feedback on the atmosphere. Coupling waves impacts on the sea surface roughness, and consequently affects wind speed
(Gentile et al., 2021), which in turn can alter the sea surface temperature.

  2) In the water column:

    a. Though surface intensified, the Stokes drift has a deeply penetrating effect throughout the Ekman layer via its interaction with the Coriolis force. This Stokes-Coriolis term can be added as an additional force in ocean module momentum equations.

b. Langmuir turbulence. This can be captured by analogy to convection, through an additional turbulent kinetic energy production term that scales with the size of the cells (Axell, 2002). Though this schemes exists in NEMO, Breivik et al. (2015) notes the mechanism is structurely different to that involving the vertical shear in the Stokes drift velocity (McWilliams et al., 1997; Polton and Belcher, 2007; Grant and Belcher, 2009) and so will have different effects on the ocean surface boundary layer.

3) At the bed:

    a. In water much shallower than the wavelength, the wave field is modified through an interaction with the bathymetry whereby the waves exert a (radiation) stress on the bed. This bottom stress can be added to ocean module's momentum equations and or wave evolution in the wave module's equations. This will often require revisiting the ocean-model's bottom friction parameters if these have been tuned in the absence of explicit wave fields.

There are choices of wave models to consider. Three state-of-the-art models are widely used: WAM (WAveModel, Komen et al. (1996)), WW3 (WAVEWATCHIII, Tolman et al. (2002)) and SWAN (Simulating WAves Nearshore, Booij et al. (1999)). They are all spectral models. Though WAM and WW3 were originally designed as deep water wave models, while SWAN was specifically built for near shore applications, they all now have capability in near shore regions. (See Cavaleri et al., 2018, for a comparative review)

As an example exploring first order interactions between the wave and ocean model in a 1.5km resolution wide area configuration of the North West European shelf the following modifications were made: 1a, 1b, 2b. Code modifications to couple WaveWatchIII to NEMO using the OASIS coupler are reported in Lewis et al. (2019a), with associated linked repositories[68].

## 4.3   Ice processes

For regional modelling in high latitudes, additional processes to represent ice are required. These include:

---

[68]https://code.metoffice.gov.uk/trac/utils/browser/ukeputils/trunk/gmd-2018


i) sea ice (frozen sea water), which prevalent in winter high latitudes and undergoes significant yearly cycling in horizontal extent and contribution to the planetary albedo.

ii) ice shelves (floating bodies of ice attached to grounded ice sheets), which regulate the flow of grounded ice towards the ocean, in turn modulating sea level rise (Scambos et al., 2004; Joughin et al., 2010) and the formation of Antarctic Bottom Water (Lewis and Perkin, 1986; Foldvik et al., 1985).

iii) icebergs (calved ice sheets), which melt and represent an important component of ice sheet mass balance and the marine stratification.

### 4.3.1 Sea ice (SI3)

Sea ice is represented using the "Sea Ice modelling Integrated Initiative" (SI3) module. SI3 is a European collaboration to pool resources and develop a unified NEMO sea ice model for regional and global applications. This module specifically targets ice

dynamics, thermodynamics, brine inclusions and subgrid-scale thickness variations, and is designed for an effective resolution of 10km or coarser. SI3 is designed in close harmony with NEMO and ships with the NEMO codebase.

Three data categories are required: ice thickness, snow thickness and ice fraction (or ice "concentration"). These are, of course, also required for the initial conditions.

Sea ice at the boundaries depend whether the ice flows into the regional domain or towards the outside. If the flow is strictly

outward, then the ice at the boundary takes the characteristics of the closest grid point inside of the domain. If the flow is inward, then boundary conditions are read from a BDY file.

Boundary data can be obtained from global simulations or reanalysis. Only three fields are required: ice fraction, and ice and snow thicknesses. Additional fields can be provided for a better accuracy: ice and snow temperature, surface temperature, ice salinity, and melt ponds concentration and thickness. If not provided, these optional fields are set to constant values determined

in the namelist. The treatment of temperature is a bit more subtle. For instance, ice and snow temperatures are derived from the surface temperature if the latter is the only field provided.

Boundary data can be single or multiple categories type. The ice code distributes data into the number of categories defined in the namelist (jpl), assuming that the ice thickness distribution follows a Gamma function as in Abraham et al. (2015). It can also distribute any number (N) of categories into jpl-categories, the only restriction being that N and jpl can only be equal

if the ice thickness distribution is the same between the parent and child models (in namelist block namitd).

On integration, the boundary fields can be set to the initial field or allow to vary, if the data exists. Only the flow relaxation (FRS) boundary condition scheme is implemented for the tracer fields, but you can use "None" if there is no sea-ice at the boundary.

At resolutions of order 10km and coarser, the continuum sea-ice model (like SI3) assumes that a grid cell contains repre-

sentative samples of different ice floes and features. At finer resolutions this assumption is weakened, and though numerically stable, the solutions at the grid scale could be less realistic.





### 4.3.2 Ice shelves and cavities

Ice shelves, their cavities, and the interactions with the ocean can be represented with a range of configuration design options.
Important processes to be considered at the design stage include: i) the role of basal melt from the underside of the shelf on the
ocean: whether this freshwater source need to be resolved or whether can it be specified (coming from observations or another
model); ii) circulation under the cavity: whether this needs to be resolved in order to improve watermass properties (Mathiot
et al., 2017), tidal processes or melt rates; iii) evolution of cavity geometry: whether the ice sheet and cavity geometry need to
evolve with time (e.g. Smith et al., 2021). These considerations influence the implementation complexity and potentially the
model stability.

In particular we focus on the representation of the ice shelf cavities, which can be open (and explicitly represented) or closed
(and therefore parameterized).

If the cavity is closed an ice melt flux can be specified. Values can be sourced from an observational study (such as Rignot
et al. (2013), Depoorter et al. (2013), Adusumilli et al. (2020)) or from an ice shelf simulation. Ideally the basal melt flux is
distributed from the base of the floating ice shelf to the grounding bathymetry. This depth range can be extracted from an ice
shelf draft data set (e.g. BedMachine Morlighem et al., 2020). A current limitation of the ice shelf module is that one can only
specify a constant melt flux.

If, on the otherhand, the cavity is open, melt rates are computed explicitly by interaction with the circulating flow. However
unrealistic melt rates can rapidly degrade the realism of a simulation. To avoid this, pay attention to the initial conditions, the
temperature and salinity biases on the continental shelf and the ice shelf module parameter settings. To minimise noisy melt
rate behaviour in a $z$-coordinate configuration, the near under-ice temperature and salinity values are averaged over a specified
boundary layer thickness.Alternatively, adopting under-ice terrain following coordinates could help alleviate instabilities by
explicitly resolving the turbulent boundary layer.

For open cavities, instead of interactively determining the melt, an option to prescribe the ice shelf basal melt in the cavities
is also available. However, it worth noting that the melt rates need to be compatible with the circulation patterns in order to
prevent unrealistic temperature and salinity values (as there is no feedback from the water temperature to the melt rate). To
build this melt rate forcing, the easiest way is to run first with interactive melt in the cavity and to extract the melt rate pattern
as your ice shelf melt forcing. Configuration options are flexible. If instabilities can not be avoided in certain cavities then a
mix of explicit and parameterized ice shelves can be employed. Finally, care must also be taken when setting the minimum
water column thickness within the cavity. To be stable, the initial water column thickness needs to be larger than the minimum
sea surface height possible (i.e. the maximal tidal range, if tides are activated, plus the magnitude of climatological sea surface
height).

### 4.3.3 Icebergs

In NEMO, icebergs can be represented in 2 different ways: by using a lagrangian model (Marsh et al., 2015) or by a climatology
(Merino et al., 2016).





Iceberg climatology: An iceberg climatology is easy to use and applicable for any regional configuration. It simply consists of a file that describes the icebergs melt for every model point. Usually, such a file is built by extracting climatological, seasonal or interannual outputs from a global simulation that uses a lagrangian iceberg model such GO6 (Storkey et al., 2018).

 Lagrangian iceberg model: Lagrangian icebergs are not compatible with regional configurations in general because they are not allowed to enter/exit across the boundaries. This being said, lagrangian icebergs can be relevant for configurations that
cover a large enough area that leads to complete melt of the icebergs before reaching the edge of the domain (e.g. a circum Antarctic simulation). In order to force the lagrangian iceberg model, the only forcing needed is a map of calving rates in $km^3$ of ice per year (ice density being fixed in the model namelist). To build this map, you need integrated calving rates per ice shelf as provided by Rignot et al. (2013), for example, plus a pattern for calving. Multiple solutions are possible to inform the spatial map of calving: all at one point per ice shelf; randomly distributed along the ice shelf front; or using output from an ice
sheet model. A tool to distribute icebergs randomly is available as part of the NEMO CDFTOOLS toolbox[69]. Finally, the user can easily define the number of categories, the size of each iceberg category and how to distribute the calved mass across the category in the model namelist. Sensitivity of results to these choices are described in Stern et al. (2016). In addition to this, it is worth keeping in mind the two major limitations of the lagrangian iceberg model in the current NEMO version: Icebergs are 'virtual', therefore they cannot serve as anchor points for land-fast ice (Li et al., 2020), and there is no fragmentation scheme
so the very large tabular icebergs do not breakup and thus have too long a lifetime (Eng, 2020).

### 4.3.4 Ice processes demonstrator

To explore a NEMO configuration with a mix of parameterized and explicit ice shelves, tides, sea ice and an iceberg climatology, a Weddell Sea worked example[70] is given in the form of a demonstrator. This is adapted from the WED025 configuration of Bull et al. (2021). The "expected results" section illustrates the melt rate and barotropic streamfunction under Filchner
Ronne Ice shelf in a 1/4° simulation. In this demonstrator the iceberg melt climatology is specified as an additional runoff source.There is no demonstrator for the lagrangian iceberg module. As reference, please see the description of the global GO6 simulation setup (Storkey et al., 2018).

## 5 Discussion and Conclusions

Following decades of groundwork development and HPC evolution, complex system regional ocean modelling is in the ascen-
dancy. The paper has two parts. In the first part the principles set out here are emergent from the current working practices and are set out in order to make the current level of endeavour sustainable. There are 3 main challenges: 1) computational oceanography is increasingly looking towards computer science to advance its capability on diverse computer architectures (Porter et al., 2018), yet the mental bandwidth of an individual human is approximately constant. 2) ocean configurations are

---

[69]https://github.com/meom-group/CDFTOOLS
[70]WED025 demonstrator: https://doi.org/10.5281/zenodo.6817000 [last accessed 22 Jul 222]





increasingly complex and hard to reproduce. 3) The research community does not have a mechanism, or is not used to, valuing
technical outputs that are contributions to the community.

We advocate a stronger implementation of Reproducible practices and make the case that through systematic workflows, and standardised assessments, skills can be democratised, debugging can be accelerated, and in general, more time can be directed to scientific questions. Containerisation could be a means to achieve some of these goals.

In this paper we have made a distinction between making a configuration reproducible (by e.g. a machine) and making the
workflow for generating a new configuration reproducible by a machine. This is because, in our scientific applications we invariably are pushing frontiers in some aspect of the configuration building process. e.g. with new process representation, grid structure, machine architecture etc. So it is desirable to accelerate the process in getting to the point of departure from standard, but not the whole process of configuration building. On the other hand, the emerging concept of Digital Twins in environmental science - purpose driven (systems of) simulations targeted at societally relevant questions - is an additional
motivation for documenting regional ocean configurations that are readily reproducible, ideally by a machine.

In this paper we have presented the case for standardised assessment, built on common tools and common diagnostics. This is motivated by the increase in data volume that routinely now requires remote, and potentially parallel, processing close to the data. In these situations the availability of proprietary software licences can not be assured and so we recommend python developing tools that are i) freely available and ii) can abstract computer science from the science by effective use of open
source packages like Dask and Xarray. We also recommend common diagnostics. Motivations here are 1) full transparency of model performance. Ideally these outputs would be made available with a configurations, whereas a paper might typically focus on more positive aspects. 2) Accelerate configuration development by having a battery of targeted tests that can be run in script form; 3) redirect duplicated effort (though there is great value in writing/understanding your own code) into more powerful share tools.

In this paper we highlight the need to recognise and value non-traditional science outputs that are increasingly an essential part of the science we do. Esteem in the traditional academic structure is gained through peer review outputs and proposal successes, but we increasingly rely on Open Source community contributions to achieve these (e.g. model source code bug fixes, enhancements and user guides or software stacks). It is our consideration that these contributions should be "cv-able" and, perhaps more challenging, that established scientists in positions of recruitment would expect to see such contributions on
CVs.

In the second part of this paper, we amass collective wisdom across a swathe of the community to offer insights on the various elements that go into building a new regional ocean configuration. These insights are targeted at new starters with the aim of passing on understanding rather than just stating what is usually done. We have distilled the advice as much as possible and abstracted details to stand-alone repositories, thereby making the advice herein more general and long lasting. In this part
we go through the build process for a realistic geometry physics configuration, then in the additional modules we introduce advice for biogeochemistry, waves, and ice processes. We do not present guidance as an instruction manual but instead aim to share the fundamentals which would equip future modellers to make strong design choices and accelerated debugging.

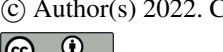



*Code and data availability.* This work was underpinned by experiences building a number regional NEMO configurations. Specifically cited in this work include:

– Workshop material featuring containerisation of an idealised NEMO regional model of Belize, to demonstrate the principles of running an ocean model and diagnosing its output (https://doi.org/10.5281/zenodo.6417227, with recipes and code), Mayorga Adame et al. (2022).

– Worked example in 1/12 degree South Asia domain (https://doi.org/10.5281/zenodo.6423211), Jardine et al. (2022)

– Worked example of the 1/12 degree South East Asia domain (https://doi.org/10.5281/zenodo.6483231), Polton et al. (2022)

– Worked example of the 500m Severn Estuary configuration (https://doi.org/10.5281/zenodo.6469990), De Dominicis et al. (2022)

– Worked example for setting up ERSEM for the SANH domain (https://doi.org/10.5281/zenodo.6907302), Partridge (2022b)

– Worked example for setting up ERSEM for the SEAsia domain (https://doi.org/10.5281/zenodo.6940832), Partridge (2022a)

– Worked example for setting up a NEMO-MEDUSA in the western Indian Ocean (https://github.com/NOC-MSM/Regional-NEMO-Medusa/ [last accessed 21Jul22])

– Worked example of Weddell Sea, Antarctica with ice capabilities (https://doi.org/10.5281/zenodo.6817000), Mathiot and Hutchinson (2022).

Other examples of documented configurations include, with less focus on How-to:

– Caribbean, (http://doi.org/10.5281/zenodo.3228088) set up with forcing data on JASMIN and provided with scripts designed to auto build and run clean configurations (Wilson et al., 2019).

– BoBEAS (https://doi.org/10.5281/zenodo.4014837) containerised and used open-MPI (Polton et al., 2020a),

– AMM7-surge (https://doi.org/10.5281/zenodo.4022310), Polton et al. (2020b)

*Author contributions.* Conceptualisation: Polton, Harle

Funding acquisition: Holt, Polton, Harle, Mayorga-Adame

Methodology: Polton, Harle

Project administration: Polton, Hutchinson

Writing - original draft preparation: Polton

Writing - review editing: All

Resources (worked examples) - Polton, Katavouta, Partridge, Jardine, Rulent, Hutchinson, De Dominicis, Mathiot, Le Guennec, Mayorga-Adame

*Competing interests.* Some authors are members of the editorial board of journal GMD. The peer-review process was guided by an independent editor, and the authors have also no other competing interests to declare





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
