# Peer review of "Reproducible and Relocatable Regional Ocean Modelling: Fundamentals and practices"

_Geoscientific Model Development, 2022_

## Author Response (AR1)

**Response to Reviewer #1 comments**

In the first part of the paper, the authors make the case for reproducible science and present a set of good practices for regional ocean modeling and present relevant software stacks. The second part goes over a comprehensive list of topics related to the setup of regional NEMO configurations (grid, bathymetry, open boundaries,...).

One could not argue against the need for more reproducibility in regional ocean models and science in general. However one important part has been overlooked and that is the reproducibility of the model solution itself. Some models (e.g. MOM6) are tested for their capabilities to reproduce answers regardless of processor layout, grid orientation, restarts and code changes. It would be interesting to detail what aspects of reproducibility are checked in the NEMO test suite and identify weaknesses, if any.

The second part details many issues that arise from setting up a regional configuration and how to handle them more or less gracefully. The authors mentioned a list of configurations previously developed and linked in the manuscript but their use in the manuscript is extremely limited, when they could have provided very valuable examples of how specific issues occur in regional domains and how they were dealt with.

Finally, the article is very NEMO focused and contains a lot of NEMO jargon that may not be obvious to the larger user community so the title should reflect that most of the content is aimed at an audience working with or interested in NEMO specifically.

Minor points:

* "bespoke" and "worked examples" are repeated many times throughout the text

We thank the reviewer for taking the time to review this manuscript. The reviewer raised three issues that we address in turn:

1.  NEMO code reproducibility.

    The reviewer highlights the important of model solution being independent of machine choice, grid orientation, code version and other "non-human" aspects. This issue is mentioned in the Troubleshooting section (3.9 line 707) but it is not the intended focus of Reproducibility section of the manuscript, which is specifically targeted on human behavioural practises to deliver (human) reproducible workflows. Nevertheless, it is an important aspect of Reproducibility in the broadest sense and its omission may cause confusion. So

we have added a comment, in the Reproducibility section, to highlight this important aspect of code development (Line 72):

"Here we specifically consider the activities required to reproduce a simulation, on the assumption that the numerical solution is independent of the discretisation implementation (e.g. machine architecture, processor decomposi- tion, grid orientation etc, which is handled by the modelling framework developers)."

2.  Specific challenges and solutions for regional domains

    The worked examples SE-Asia, SANH and SEVERN-SWOT are highlighted as "key" worked examples. The associated documentation (wiki on the repositories or the PDFs in the zenodo link) provide exhaustive detail on how to resolve the specific issues that arise with these specific domains, as the reviewer request. Earlier drafts of the manuscript included the region specific treatment of issues but the document became unwieldy (and perhaps of less broad interest) so we instead focussed on concise and generic advice in the main manuscript and reserved region specific detail to the worked example "assets". Our sign-posting of this separation was perhaps not clear enough. This has been addressed in the revision. For example, the abstract states:

    **"Detail and region specific worked examples are linked in companion repositories and DOIs"**

3.  NEMO focus not sign-posted in the title

    We acknowledge that the manuscript includes NEMO applications of the principles presented. However this advice is not NEMO specific but general to regional oceanography (albeit hydrostatic ocean modelling). In particular, we believe that the fundamental concepts are of interest to a broader readership which would be needlessly limited by making the title too specialising. We have worked hard to include many worked examples to make this document more useful – these naturally are NEMO specific as we are NEMO users. But we don't believe that turns this into a manuscript only for NEMO users. The application to NEMO is clearly stated in the abstract and the manuscript is submitted to the NEMO special issue. In this way we hope that the connection to NEMO is not overlooked or misinterpreted.

Minor comments:

*   We have reviewed the text and reigned in excessive use of phrases "bespoke" and "worked examples". We agree that use of "bespoke" was out of control and heavily edited its use. However use of "worked example" in general

pointed towards the external material which highlighted relevant and specific applications to the general insights shared (which was raised as an issue: point 2). So we did less to curtail this usage.

**Response to Reviewer #2 comments**

*These two sections built a great deal of expectation for the three worked examples. The concept of working examples that researchers can follow along through steps as they build their regional domain is excellent. They can try the method on the example, test that they can replicate it before they try their own. This process separates method problems from local problems: wonderful.*

*However, when I examined one of the examples: SEVERN-SWOT I was disappointed. As someone not having access to the ARCHER2 machine, it was not possible for me to set up and run this example.*

*E.g. 1) Reading the pdf[1], we quickly come to making the new regional model domain from the larger AMM15 domain. The AMM15 coordinates file location is given as a directory on archer2. I know I can get this file from the NEMO site (or could in the past) but new people will not necessarily know this.*

*E.g. 2) As the Severn is an estuary I was particularly interested in the river forcing. There is nothing about the river forcing in the pdf but I did find a README[2] under inputs/rivers on the github page. However, again, it points to a file on archer and I do not know another source for it.*

*Not only can I not make the SEVERN-SWOT regional example following along with the pdf, I don't think I could even get a working final SEVERN-SWOT model going.*

*I know the authors intend for their models to be reproducible beyond the boundaries of their computer system. However, it's always the details and generally the big input/forcing files where the intention meets reality. Here, however, as a demonstration of the principle, I had hoped to find the exception. I don't think it would be hugely difficult to make this example accessible to the greater audience. It will take careful review and perhaps putting some of the larger files on a website.*

We thank the reviewer for taking the time to review this manuscript and are pleased that they saw it as valuable, especially for new modellers.

Issue about availability of example files

This is a fair criticism. Though we are not aiming to present a "black box" solution that is machine independent, we are trying to make it easier for people to follow along. Having critical files unnecessarily unobtainable rather defeats this aspiration! We have carefully gone through the Severn Estuary configuration documentation, in particular, and identified the fundamental files required to independently produce the workflow. There are two types of file: 1) files we "own" e.g. the AMM15 coordinates file, and 2) files we do not own. E.g. ERA5 met forcing, FES2014 tidal files, open boundary conditions from the CMEMS catalogue, GEBCO bathymetry and river forcing data. For the former (coordinates file) we attach the binary file to the updated Zenodo repository: https://zenodo.org/record/7473198. For the latter we have updated the wiki, where previously it only had links to internal storage with urls for the download sites. Documentation for the Indian subcontinent and SE Asia domains already had links to the external file locations and the coordinates file was similarly already hyperlinked.

The worked examples presented in the paper are not necessarily static (though the DOIs point to specific releases). Recent development in the SEVERN-SWOT case study includes modifications to run with wetting & drying and river forcing. In order to include river forcing as a worked example, we have followed and linked the methodology used in the SEAsia example. Although, this might not be the best river forcing for the Severn, it shows the methodology to be used, and the data can easily be switched if a user has a preferred source.

***Details*:**

- Line 257 xarray repeated

Done

- Line 248 workflows…. that abstract  (no s on abstract)

Done

- Line 405 important to ensure that straits are not connected on the diagonal only (no flow that way) and that important islands have not been remove.  Using (and keeping) a script (even a complicated one) to manipulate your bathymetry means that it is reproducible.

Agreed. There is an example in the SEVERN-SWOT documentation. So this is pointed to.

- Line 562 such as biogeochemistry  (s missing on as)

Done

- Line 605 spell out MJO

Done

- Line 633 10 m is very deep for mixing freshwater in my coastal experience.  Plumes (Rhine, Columbia etc) are not 10 m thick

Agreed. Using a 10m mixing depth was a pragmatic response to stability issues when using biogeochemical and physical river variables. We confess to never getting to the bottom of this issue and are cautious about interpretation of simulation near the coastline. We have amended the text to make this clearer:

"In all our mid latitude and tropical applications with biogeochemistry we mixed the freshwater over the top 10m, for numerical stability."

- Line 704 input files (no s on input)

Done

- Line 730 in strong tidal mixing areas, with good vertical resolution, the vertical CFL number can also be a problem

Agreed. This is easily overlooked. A caution about vertical CFL criteria is added for shallow tidal regimes.

- Line 916 this scheme exists (no s on scheme)

Done

- Line 957 and on: this level of detail, mentioning specific variables, is much higher than other sections of the paper, I suggest abstracting it to match the rest of the paper.

Agreed. Removed lines making specific mention of parameter names. These are introduced more thoroughly in the linked WED025 Demonstrator.

- Line 970 source needs (s on need)

Done

- Line 998 and on: Lagrangian (with a capital as Lagrange was a person)

 Done

- General quibble: The paper stresses consistent boundary conditions, river forcing, atmospheric forcing.  I agree that consistent helps avoid some bizarre errors. However, the coastal ocean is very much a receiver of forcings and accurate forcing can be really key for some processes in any given region.

We press the point about consistency only in the river section. However, the point that data quality may trump data consistency for some processes is more generally true. For example if

biogeochemical data boundary conditions are required then physics plus biogeochemisty boundary data will almost certainly come only be available at a lower resolution compared to sourcing physics-only boundary conditions.

Because this advice is more generally true we have revised the text accordingly line 629:

"However, while consistent forcing is desirable, a dataset with a range of consistent variables may be lower accuracy than e.g. a region specific flow only dataset. In some strongly forced applications, forcing accuracy in specific variables maybe more important than consistency across all forcing variables. See SANH[56] for an example that generates river forcing from different sources, and section 4.1.3 for specific guidance on constructing riverine biogeochemical fluxes."

---

## Author Response (AR2)

**Response to Editor comments**

The authors thank the editors for their gracious handling of the monster manuscript.

By Olivier Marti

> Dear authors,
>
> I thank you for this revised version and your convincing responses to the reviewers.
>
> The paper is almost ready, but I still have a concern about the title.
>
> In my first comment, I wrote "The paper address some general issues about regional ocean modeling. But a large part of the paper is based on applications and example with NEMO. So I think that is better, considering the general policies of GMD, to put a reference to NEMO in the title. "I think" means that I'm open to discussion if your opinion differs."
>
> Then reviewers #1 writes comment #3 "NEMO focus not sign-posted in the title". You answered that the paper is of general scope and that you wanted to avoid to mention NEMO in the title. And that the abstract clearly explains that.
>
> I find the abstract a bit misleading on this subject : "**This advice is compiled from across the user community, is presented in the context of NEMOv4, though aims to transcend NEMO version**." With this formulation, the reader may understand that to "transcend NEMO version" means that other NEMO version than v4 (older or future ones) might be concerned. If the project of the paper is to encompass regional ocean modelling with any model, this phrase does not reflect it, and needs a rephrasing.
>
> Best regards, and thank you for publishing in GMD.
>
> Olivier Marti
> GMD Topical Editor

We are grateful for the careful reading and help we have had getting this right. We have revised the misleading section in the abstract accordingly to both clarify that we offer general advice for any regional ocean model, though we limit the practical examples to NEMOv4. This section now reads:

This advice is compiled from across the NEMO user community and sets out principles and practises that encompass regional ocean modelling with any model. Detail and region specific worked examples in Sections 3, 4, the linked companion repositories and DOIs all target NEMOv4.

By Una Miškovic

1. Coloured or marked text in *.pdf manuscript file is not allowed. Please provide a clean version of *pdf manuscript file (with black text) for the next revision.

Done

2. For the next revision, I kindly ask you to remove the placeholder "Copyright statement" from page #1 in *.pdf manuscript version.

Done

3. Please avoid the use of footnotes. They have to be implemented within the main text, if possible.

Done. All the footnotes have been removed and the text has been moved into the body of the manuscript. (As an aside, I do not think that the loss of footnotes improves readability in away way but understand it could be a requirement to manage conversion to markdown. If there is a wishlist for system development, I would add better support footnotes).